# 🪞 TokUR: Token-Level Uncertainty Estimation for Large Language Model Reasoning

**Tunyu Zhang**[*†1] **Haizhou Shi**[*†1] **Yibin Wang**[2] **Hengyi Wang**[1] **Xiaoxiao He**[1] **Zhuowei Li**[1]
**Haoxian Chen**[‡3] **Ligong Han**[4] **Kai Xu**[4] **Huan Zhang**[2] **Dimitris Metaxas**[1] **Hao Wang**[†1]
[1]Rutgers University  [2]UIUC  [3]Amazon  [4]Red Hat AI Innovation

## Abstract

While Large Language Models (LLMs) have demonstrated impressive capabilities, their output quality remains inconsistent across various application scenarios, making it difficult to identify trustworthy responses, especially in complex tasks requiring multi-step reasoning. In this paper, we propose a **Tok**en-level **U**ncertainty estimation framework for **R**easoning (**TokUR**) that enables LLMs to self-assess and self-improve their responses in mathematical reasoning. Specifically, we introduce low-rank random weight perturbation during LLM decoding to generate predictive distributions for token-level uncertainty estimation, and we aggregate these uncertainty quantities to capture the semantic uncertainty of generated responses. Experiments on mathematical reasoning datasets of varying difficulty demonstrate that `TokUR` exhibits a strong correlation with answer correctness and model robustness, and the uncertainty signals produced by `TokUR` can be leveraged to enhance the model's reasoning performance at test time. These results highlight the effectiveness of `TokUR` as a principled and scalable approach for improving the reliability and interpretability of LLMs in challenging reasoning tasks. The source code is avaliable at https://github.com/Wang-ML-Lab/TokUR.

## 1 Introduction

Large Language Models (LLMs) have demonstrated remarkable capabilities in various reasoning tasks (Wei et al., 2022a; Wang et al., 2022; Chung et al., 2024; Guo et al., 2025), yet they often struggle to reliably assess the quality of their own responses (Xiong et al., 2023; Tian et al., 2023; Kapoor et al., 2024; Liu et al., 2024; Zhang & Zhang, 2025; Da et al., 2025; Liu et al., 2025). This limitation becomes particularly evident in complex reasoning scenarios where models may generate seemingly convincing but incorrect solutions without indicating uncertainty.

Beyond the dominant body of uncertainty estimation methods that largely focus on short-form question answering (Zhang et al., 2023; Yadkori et al., 2024) and classification tasks (Yang et al., 2023; Wang et al., 2024; Shi et al., 2024), two main approaches have been explored for the more challenging setting of sequence uncertainty estimation: **(i) Query-level methods** (Gal et al., 2016; Osband et al., 2023; Hou et al., 2023), despite their solid theoretical foundation, estimate uncertainties $\mathcal{U}(\mathbf{y}|\boldsymbol{x})$ with respect to *input prompts $\boldsymbol{x}$ alone*, without evaluating *the quality of specific generated responses $\boldsymbol{y}$* conditioned on those inputs (see Sec. 2.1). Besides, these methods require marginalization over the entire output space $\mathbf{y}$; this becomes intractable as sequence length grows. **(ii) Response-level methods** (Murray & Chiang, 2018; Malinin & Gales, 2021; Kadavath et al., 2022), typically variants of log-probabilities, have shown empirical success but lack strong theoretical grounding (Kuhn et al., 2023). As a result, the limitations of the aforementioned methods in capturing response-specific uncertainty hinder the deployment of LLMs in high-stakes reasoning tasks that demand reliable self-assessment.

To address this challenge, we propose a principled framework, dubbed **Tok**en-level **U**ncertainty estimation for **R**easoning (TokUR), for estimating the uncertainty of generated sequences by aggregating token-level uncertainties based on random low-rank weight perturbation. TokUR introduces carefully

---

[*]Equal Contribution. [†]Correspondence to: Tunyu Zhang <ty.zhang@rutgers.edu>, Haizhou Shi <haizhou.shi@rutgers.edu>, Hao Wang <hw488@cs.rutgers.edu>. [‡]Work done outside Amazon.

calibrated perturbations to the weights of attention layers, creating an ensemble of model variants that enables principled uncertainty estimation without requiring costly retraining or extensive parameter updates. Building on this, we decompose the *total uncertainty* of each generated token into *aleatoric uncertainty* (inherent randomness in the data) and *epistemic uncertainty* (model uncertainty about its parameters), providing a theoretically grounded assessment of confidence across the generation process. We then aggregate these token-level uncertainties to evaluate entire reasoning responses, demonstrating both theoretical consistency with established uncertainty principles and practical utility in downstream applications.

Empirically, `TokUR` enhances LLM reasoning in three key aspects: (i) token-level *epistemic uncertainty* effectively identifies incorrect reasoning paths, outperforming baselines across three mathematical reasoning benchmarks, (ii) `TokUR` excels at selecting high-quality solutions from multiple candidates, and (iii) it functions as an implicit reward to guide reasoning, improving accuracy when combined with off-the-shelf test-time-scaling algorithms (Puri et al., 2025). In summary, our contributions are:

- We introduce `TokUR`, a training-free token-level uncertainty estimation approach for LLM reasoning through low-rank weight perturbation, providing a principled decomposition of uncertainties with proven theoretical properties.
- We demonstrate that epistemic uncertainty can serve as a good metric to measure the quality of generated reasoning paths, consistently outperforming conventional confidence metrics across diverse mathematical reasoning tasks.
- We demonstrate practical applications[1] of our uncertainty estimation framework: it improves reasoning performance through incorrect path detection, high-quality solution selection, and uncertainty-guided generation.

## 2 PRELIMINARIES

In this section, we first introduce the notation used in the remaining sections, and then review the key concepts of uncertainties (Sec. 2.1) and existing Bayesian LLMs for downstream adaptation (Sec. 2.2).

**Notation.** In this paper, scalars are denoted by lowercase letters ($x$), vectors by lowercase bold-math letters ($\boldsymbol{x}$), random vectors by lowercase boldface letters ($\mathbf{x}$), and matrices by uppercase boldface letters ($\boldsymbol{X}$). We use $[m] = \{1, 2, \cdots, m\}$ to denote the set of consecutive integer numbers from $1$ to $m$. Following convention, we use $p$ for probability, $\mathbb{E}$ for expectation, $\mathcal{H}$ for entropy, and $\mathcal{I}$ for mutual information. Specifically, $\mathcal{H}[\mathbf{y}|\mathbf{x}]$ denotes the conditional entropy between *random variables* $\mathbf{y}$ and $\mathbf{x}$. We use $\mathcal{H}[p(\mathbf{y}|\mathbf{x} = \boldsymbol{x})]$ to denote the *predictive entropy* of the output variable conditioned on input $\boldsymbol{x}$, with $\mathcal{H}[p(\mathbf{y}|\boldsymbol{x})]$ as a shorthand notation when context is clear.

### 2.1 UNCERTAINTY ESTIMATION OF LONG-FORM GENERATION

**Prediction with Bayesian Neural Networks.** Bayesian Neural Networks (BNNs) (Neal, 2012; Hernández-Lobato & Adams, 2015; Gal & Ghahramani, 2016; Blundell et al., 2015; Wang & Yeung, 2016; Wang et al., 2016; Lakshminarayanan et al., 2017; Wang & Yeung, 2020) predict responses and estimate their uncertainties using the variational distribution $q(\boldsymbol{\theta}|\mathcal{D})$ that approximates the true weight posterior $p(\boldsymbol{\theta}|\mathcal{D})$. Given an input sequence $\boldsymbol{x} = (x_1, \cdots, x_L) \in \mathcal{X}$, the probability of the output sequence $\boldsymbol{y} = (y_1, \cdots, y_T) \in \mathcal{Y}$ is defined as marginalization over the parameters and estimated by Bayesian Model Averaging (BMA) of size $M$:

$$p(\boldsymbol{y}|\boldsymbol{x}) = \int p(\boldsymbol{y}|\boldsymbol{x}; \boldsymbol{\theta})\, q(\boldsymbol{\theta}|\mathcal{D})\, d\boldsymbol{\theta} \approx \frac{1}{M} \sum_{m=1}^{M} p(\boldsymbol{y}|\boldsymbol{x}; \boldsymbol{\theta}^{(m)}), \quad \boldsymbol{\theta}^{(m)} \sim q(\boldsymbol{\theta}|\mathcal{D}). \quad (1)$$

**Query-Level Uncertainty Estimation.** Established techniques of uncertainty estimation (Gal et al., 2016) mainly quantify the uncertainty of input $\boldsymbol{x}$ (*query-level uncertainty*) by

$$\mathcal{H}[p(\mathbf{y}|\boldsymbol{x})] = \mathbb{E}_{\boldsymbol{y} \sim p(\mathbf{y}|\boldsymbol{x})}[-\log p(\boldsymbol{y}|\boldsymbol{x})]. \quad (2)$$

---

[1] We provide an implementation of our framework that is compatible with vLLM (Kwon et al., 2023) for efficient deployment.

In the context of BNNs (Eqn. 1), the predictive distribution of $\boldsymbol{y}$ is the marginalized predictive distribution over the model parameters, and hence Eqn. 2 is defined as *"total uncertainty"* (Gal et al., 2016; Depeweg et al., 2017).

A model's uncertainty about a specific input cannot be solely attributed to the randomness of the approximate posterior $q(\boldsymbol{\theta}|\mathcal{D})$, which is input-agnostic. For instance, when faced with a query *"Name a city in the UK?"* (Yadkori et al., 2024), even if an infinite amount of data is observed (eliminating the randomness of the model parameters), the uncertainty of this question remains high, as there are many correct candidate answers. Hence to distinguish different sources of uncertainty, *total uncertainty* is decomposed into *epistemic uncertainty* and *aleatoric uncertainty* (Gal et al., 2016):

$$\underbrace{\mathcal{H}[p(\mathbf{y}|\boldsymbol{x})]}_{\text{Total Uncertainty}} = \underbrace{\mathbb{E}_{q(\boldsymbol{\theta}|\mathcal{D})}[\mathcal{H}[p(\mathbf{y}|\boldsymbol{x};\boldsymbol{\theta})]]}_{\text{Aleatoric Uncertainty}} + \underbrace{\mathcal{I}(\mathbf{y};\boldsymbol{\theta}|\boldsymbol{x})}_{\text{Epistemic Uncertainty}}. \tag{3}$$

Here, *aleatoric uncertainty* captures the intrinsic randomness in data and cannot be reduced even with more data observed. In contrast, *epistemic uncertainty*, defined as the mutual information $\mathcal{I}(\mathbf{y};\boldsymbol{\theta}|\boldsymbol{x})$ between $\mathbf{y}$ and $\boldsymbol{\theta}$, reflects the model's uncertainty about its own parameters, which can in principle be reduced by collecting more evidence. We use $\mathcal{U}(\mathbf{y}|\boldsymbol{x})$ defined in Definition 2.1 to denote any of the three uncertainties.

**Definition 2.1** (**Query-Level Uncertainty**). *Query-level uncertainty $\mathcal{U}(\mathbf{y}|\boldsymbol{x})$ is the uncertainty of the predictive distribution $p(\mathbf{y}|\boldsymbol{x})$ given an input query $\boldsymbol{x}$. Total Uncertainty (TU), Aleatoric Uncertainty (AU), and Epistemic Uncertainty (EU) in Eqn. 3 are all instances of query-level uncertainty.*

**Limitations of Query-Level Uncertainty.** Using the chain rule for conditional entropy (Cover, 1999), the *query-level uncertainty* estimation can be decomposed token-by-token as

$$\mathcal{U}(\mathbf{y}|\boldsymbol{x}) = \sum_{t=1}^{T} \mathcal{U}(\mathbf{y}_t|\mathbf{y}_{<t},\boldsymbol{x}). \tag{4}$$

However, the uncertainty term $\mathcal{U}(\mathbf{y}_t|\mathbf{y}_{<t},\boldsymbol{x})$ in Eqn. 4 requires marginalization over the random variable $\mathbf{y}_{<t}$, which is *(i) computationally intractable,* and *(ii) only reflecting the quality of the input query.* Hence, these query-level uncertainties are not proper indicators for evaluating a concrete output response $\boldsymbol{y}$.

## 2.2 BAYESIAN LARGE LANGUAGE MODELS

**Bayesian Low-Rank Adaptation.** For a pre-trained network layer with weight matrix $\boldsymbol{W}_0$, Low-Rank Adaptation (LoRA) (Hu et al., 2022) optimizes the parameters within a constrained low-rank subspace. Specifically, the weight update matrix is modeled by $\Delta\boldsymbol{W} = \boldsymbol{B}\boldsymbol{A}$, where $\Delta\boldsymbol{W} \in \mathbb{R}^{m \times n}$, $\boldsymbol{B} \in \mathbb{R}^{m \times r}$, $\boldsymbol{A} \in \mathbb{R}^{r \times n}$, and $r \ll \min(m,n)$. The output $\boldsymbol{z} \in \mathbb{R}^{m \times 1}$ of forwarding the input vector $\boldsymbol{h} \in \mathbb{R}^{n \times 1}$ is then

$$\boldsymbol{z} = \boldsymbol{W}_0\boldsymbol{h} + \Delta\boldsymbol{W}\boldsymbol{h} = \boldsymbol{W}_0\boldsymbol{h} + \boldsymbol{B}\boldsymbol{A}\boldsymbol{h}. \tag{5}$$

Leveraging LoRA's parameter efficiency, Bayesian LoRAs (Yang et al., 2023; Wang et al., 2024; Shi et al., 2024) aim to further integrate BNN's uncertainty estimation capabilities into LLMs without significant increasing memory complexity. The key idea is to model $\boldsymbol{A}$ and/or $\boldsymbol{B}$ as approximate distributions of the true weight posterior. The asymmetric Bayesianization, exemplified by BLoB (Wang et al., 2024) and TFB (Shi et al., 2024), models the elements of $\boldsymbol{A}$ with independent Gaussian distributions while keeping $\boldsymbol{B}$ deterministic. Specifically, we have

$$q(\boldsymbol{A}|\{\boldsymbol{M},\boldsymbol{\Omega}\}) = \prod_{ij} q(A_{ij}|M_{ij},\Omega_{ij}) = \prod_{ij} \mathcal{N}(A_{ij}|M_{ij},\Omega_{ij}^2), \tag{6}$$

where $\boldsymbol{M}$ and $\boldsymbol{\Omega}$ share the same shape as $\boldsymbol{A}$ and denote the mean and standard deviation of the random variable $\boldsymbol{A}$, respectively. To estimate this distribution, BLoB jointly trains the mean and covariance through the re-parameterization trick (Wang et al., 2024), while TFB uses a simple training-free maximal variance searching technique by fixing the approximate distribution to the family of low-rank isotropic Gaussian distributions (Shi et al., 2024).

**Limited Scope of Existing Bayesian LLMs.** Existing Bayesian LLMs have been primarily validated in downstream classification tasks of simple single- or multiple-choice problems, where uncertainty estimation is quantitatively assessed via the alignment of prediction confidence and

accuracy (Yang et al., 2023; Balabanov & Linander, 2024; Wang et al., 2023; 2024; Shi et al., 2024). However, these methods have not yet demonstrated effective generalization to long-form generation tasks, i.e., LLM reasoning. Therefore, our TokUR, which estimates token-level uncertainties via weight perturbations, represents an initial step toward *extending Bayesian LLMs to long-form generation*, an area where uncertainty estimation remains largely unexplored and technically challenging.

## 3 TokUR: Token-Level Uncertainty Estimation via Low-Rank Weight Perturbation

Sec. 3.1 introduces the key techniques of token-level uncertainty estimation. Sec. 3.2 then details how token-level uncertainties can be aggregated for response-level uncertainty estimation, and describes the underlying theoretical foundation. Finally, Sec. 3.3 presents our low-rank weight perturbation as posterior approximation. **All proofs of propositions can be found in Appendix C.**

### 3.1 Token-Level Uncertainties in General

Given an approximate posterior $q(\boldsymbol{\theta}|\mathcal{D})$, a fixed input query $\boldsymbol{x} \in \mathcal{X}$ and a *specific output response* $\boldsymbol{y} = (y_1, y_2, \ldots, y_T) \in \mathcal{Y}$ sampled from the base policy $p(\boldsymbol{y}|\boldsymbol{x})$, we denote the predictive distribution of the next token $y_t$ produced by marginalization over weights as

$$\bar{p}(y_t|\boldsymbol{y}_{<t}, \boldsymbol{x}) \triangleq \mathbb{E}_{\boldsymbol{\theta} \sim q(\cdot|\mathcal{D})}[p(y_t|\boldsymbol{y}_{<t}, \boldsymbol{x}; \boldsymbol{\theta})]. \tag{7}$$

**Assumption 3.1** (**Stepwise Posterior Sampling**). *We assume that the weights $\boldsymbol{\theta}$ sampled from the approximate posterior $q(\cdot|\mathcal{D})$ are* not shared *across decoding steps. Formally, the probability of a sequence is factorized as*

$$\bar{p}(\boldsymbol{y}|\boldsymbol{x}) \triangleq \prod_{t=1}^{T} \bar{p}(y_t|\boldsymbol{x}, \boldsymbol{y}_{<t}) = \prod_{t=1}^{T} \left\{ \mathbb{E}_{\boldsymbol{\theta}_t \sim q(\cdot|\mathcal{D})}[p(y_t|\boldsymbol{x}, \boldsymbol{y}_{<t}, \boldsymbol{\theta}_t)] \right\}, \tag{8}$$

*instead of adopting the joint formulation*

$$\bar{p}(\boldsymbol{y}|\boldsymbol{x}) \triangleq \mathbb{E}_{\boldsymbol{\theta} \sim q(\cdot|\mathcal{D})}[p(\boldsymbol{y}|\boldsymbol{x}, \boldsymbol{\theta})]. \tag{9}$$

While both are valid probabilistic models, the joint formulation is incompatible with the autoregressive decoding mechanism of LLMs. Hence, we adopt the stepwise formulation in Assumption 3.1. To validate this assumption, we further conduct an ablation study comparing the stepwise formulation in Assumption 3.1 with the joint formulation, and report the results in Appendix E.5.4.

Given an input $\boldsymbol{x}$ and a partial output $\boldsymbol{y}_{<t}$, for the time step $t$, we have the following three uncertainties:

- **Total Uncertainty (TU)** is the entropy of random variable $\mathbf{y}_t$ conditioned on $\boldsymbol{x}$ and $\boldsymbol{y}_{<t}$:

$$\text{TU}(\mathbf{y}_t|\boldsymbol{y}_{<t}, \boldsymbol{x}) \triangleq \mathcal{H}[\bar{p}(\mathbf{y}_t|\boldsymbol{y}_{<t}, \boldsymbol{x})] = -\sum_{y_t \in \mathcal{V}} \bar{p}(y_t|\boldsymbol{y}_{<t}, \boldsymbol{x}) \log \bar{p}(y_t|\boldsymbol{y}_{<t}, \boldsymbol{x}), \tag{10}$$

- **Aleatoric Uncertainty (AU)** is the expectation of entropy of random variable $\mathbf{y}_t$ over the weights $\boldsymbol{\theta}$ sampled from the approximate posterior $q(\cdot|\mathcal{D})$ as in Eqn. 3:

$$\text{AU}(\mathbf{y}_t|\boldsymbol{y}_{<t}, \boldsymbol{x}) \triangleq \mathbb{E}_{\boldsymbol{\theta} \sim q(\cdot|\mathcal{D})}\left[\mathcal{H}[p(\mathbf{y}_t|\boldsymbol{y}_{<t}, \boldsymbol{x}; \boldsymbol{\theta})]\right], \tag{11}$$

- **Epistemic Uncertainty (EU)** is the difference between TU and AU:

$$\text{EU}(\mathbf{y}_t|\boldsymbol{y}_{<t}, \boldsymbol{x}) \triangleq \text{TU}(\mathbf{y}_t|\boldsymbol{y}_{<t}, \boldsymbol{x}) - \text{AU}(\mathbf{y}_t|\boldsymbol{y}_{<t}, \boldsymbol{x}) = \mathcal{I}(\mathbf{y}_t; \boldsymbol{\theta}|\boldsymbol{y}_{<t}, \boldsymbol{x}), \tag{12}$$

where $\mathcal{V}$ is the vocabulary and all the expectations are estimated with BMA.

### 3.2 Token-Level Uncertainty for Response-Level Uncertainty Estimation

**Definition 3.1** (**Response-Level Uncertainty**). *Given the token-level uncertainties $\mathcal{U}(\mathbf{y}_t|\boldsymbol{y}_{<t}, \boldsymbol{x})$ defined in Eqn. 10-12, we define* response-level uncertainty *as their cumulative sum across all tokens in the output sequence:*

$$\widetilde{\mathcal{U}}(\boldsymbol{y}|\boldsymbol{x}) \triangleq \sum_{t=1}^{T} \mathcal{U}(\mathbf{y}_t|\boldsymbol{y}_{<t}, \boldsymbol{x}), \tag{13}$$

*where $\mathcal{U}$ can denote any of the considered uncertainty measures (TU, AU, or EU in Eqn. 10-12).*

**Proposition 3.1** (**Response-Level Uncertainty as an Unbiased Estimator of Query-Level Uncertainty**). *Given an input query $\boldsymbol{x}$, let $\boldsymbol{y} \sim p(\mathbf{y}|\boldsymbol{x})$ be a generated sample of length $T$. Then the response-level uncertainty $\widetilde{\mathcal{U}}$ (Definition 3.1) is an **unbiased estimator** of the query-level uncertainty $\mathcal{U}$ (Definition 2.1), i.e.,*

$$\mathbb{E}_{\boldsymbol{y}\sim p(\mathbf{y}|\boldsymbol{x})}[\widetilde{\mathcal{U}}(\boldsymbol{y}|\boldsymbol{x})] = \mathcal{U}(\mathbf{y}|\boldsymbol{x}). \tag{14}$$

**Proposition 3.2** (**Token-Level and Response-Level Uncertainty**). *Given an input query $\boldsymbol{x}$, let $\boldsymbol{y} \sim p(\mathbf{y}|\boldsymbol{x})$ be a generated sample of length $T$. Let $\mathcal{U}(\mathbf{y}_t|\boldsymbol{y}_{<t},\boldsymbol{x})$ denote the token-level uncertainty as defined in Eqn. 10-12, with $\widetilde{\mathcal{U}}(\mathbf{y}|\boldsymbol{x})$ as the corresponding response-level uncertainty (Definition 3.1). Our token-level uncertainty is equivalent to the response-level uncertainty when $T = 1$:*

$$\widetilde{\mathcal{U}}(\boldsymbol{y}_1|\boldsymbol{x}) = \mathcal{U}(\mathbf{y}_1|\boldsymbol{x}). \tag{15}$$

The two propositions above provide key connections between Eqn. 13 and existing uncertainty estimation theory (Malinin & Gales, 2021; Ling et al., 2024). Proposition 3.1 shows that $\widetilde{\mathcal{U}}(\mathbf{y}|\boldsymbol{x})$ is an unbiased estimator of the true query-level uncertainty $\mathcal{U}(\mathbf{y}|\boldsymbol{x})$, ensuring its statistical consistency with the ideal formulation. Proposition 3.2 confirms that when the sequence length $T = 1$, e.g., single-token prediction tasks such as multiple-choice QA (Yang et al., 2023; Wang et al., 2024), the estimator exactly recovers the token-level uncertainty, demonstrating structural consistency. These results support the validity and reliability of our approximation.

**Advantages of Token-Level Uncertainty.** Compared to Query-Level Uncertainty (Definition 2.1), token- and response-level uncertainties *(i) avoid expensive marginalization over sequences* (note the difference between $\mathbf{y}_{<t}$ in Eqn. 4 and $\boldsymbol{y}_{<t}$ in Eqn. 14) while still *(ii) capturing the expected uncertainty conditioned on the generated output response.* Moreover, since $\mathcal{U}(\mathbf{y}_t|\boldsymbol{y}_{<t},\boldsymbol{x})$ depends on the quality of the prefix $\boldsymbol{y}_{<t}$, *(iii) the estimate retains rich semantic information,* making it well-suited for entropy-based sequential decision-making (Kuhn et al., 2023; Ye et al., 2025) or hallucination detection (Farquhar et al., 2024; Kossen et al., 2024; Ye et al., 2025) in downstream tasks.

### 3.3 Low-Rank Weight Perturbation as Approximation of Weight Posterior

Suppose that we have an LLM policy $p(\mathbf{y}|\mathbf{x})$. To estimate the uncertainty of its output, we cast this model into a Bayesian framework by introducing weight perturbations. Due to the established advantages of *efficiency*, *performance preservation* of pre-perturbation model, and *effectiveness* of uncertainty estimation (Shi et al., 2024), we adopt a low-rank structure for the noise added to the model weights. Given a rank-$r$ weight matrix $\boldsymbol{W}_0 \in \mathbb{R}^{m\times n}$ of a neural network layer, we first perform compact Singular Value Decomposition (SVD) (Klema & Laub, 1980):

$$\boldsymbol{W}_0 = \boldsymbol{U}\,\mathrm{diag}(\boldsymbol{d})\boldsymbol{V}^{\top}, \tag{16}$$

where $\boldsymbol{d} \succ \boldsymbol{0} \in \mathbb{R}^{r\times 1}$ is the vector of singular values, and $\boldsymbol{U} \in \mathbb{R}^{m\times r}$ and $\boldsymbol{V} \in \mathbb{R}^{n\times r}$ both contain orthonormal columns, i.e., $\boldsymbol{U}^{\top}\boldsymbol{U} = \boldsymbol{V}^{\top}\boldsymbol{V} = \boldsymbol{I}_r$. To ensure computational efficiency, we introduce a low-rank noise matrix $\boldsymbol{\epsilon} \in \mathbb{R}^{n\times r'}$ whose rank $r' \ll r$ is significantly smaller than the rank of weight matrix, and whose entries are sampled i.i.d. from a Gaussian distribution of standard deviation of $\sigma_q$, which we refer to as *perturbation strength*, i.e., $\epsilon_{ij} \sim \mathcal{N}(0, \sigma_q^2), \forall i \in [n], j \in [r']$. The perturbed weight matrix is then constructed as

$$\boldsymbol{W} = \boldsymbol{W}_0 + \boldsymbol{U}'\boldsymbol{\epsilon}^{\top}, \tag{17}$$

where the matrix $\boldsymbol{U}'$ contains the top-$r'$ columns of $\boldsymbol{U}$. This perturbation transforms the deterministic $\boldsymbol{W}_0$ to a variational low-rank isotropic Gaussian distribution $\boldsymbol{W}$ (Wang et al., 2024; Shi et al., 2024):

$$\begin{aligned}
q(\mathrm{vec}(\boldsymbol{W})|\sigma_q) &= \mathcal{N}(\mathrm{vec}(\boldsymbol{W})|\boldsymbol{\mu}_q, \boldsymbol{\Sigma}_q), \\
\text{where} \quad \boldsymbol{\mu}_q &= \mathrm{vec}(\boldsymbol{W}_0), \\
\boldsymbol{\Sigma}_q &= \sigma_q^2 \boldsymbol{I}_n \otimes \begin{bmatrix} \boldsymbol{I}_{r'} & \\ & \boldsymbol{0}_{m-r'} \end{bmatrix}.
\end{aligned} \tag{18}$$

Let $\boldsymbol{\theta}$ denote the collection of all perturbed weight matrices across the model. By assuming the statistical independence among layers, the overall approximate posterior becomes

$$q(\boldsymbol{\theta}|\sigma_q) = \prod_i q(\mathrm{vec}(\boldsymbol{W}^i)|\sigma_q). \tag{19}$$

Figure 1: **Distribution of `TokUR`'s Uncertainty Scores and AUROC across Different Difficulty Levels,** applied to `Llama-3.2-1B-Instruct`. **Left:** `TokUR` (AU, Ours); **Middle:** `TokUR` (TU, Ours); **Right:** `TokUR` (EU, Ours).

**Utilizing the Approximate Weight Posterior** $q(\boldsymbol{\theta}|\sigma_q)$. Notably, while we leverage the variational posterior formulation of Eqn. 19 to quantify uncertainty (detailed in Sec. 3.1), we use only the mean weights $\boldsymbol{W}_0$ for decoding of each step rather than BMA as in Eqn. 1. This approach allows for a controlled study of the effects of uncertainty estimation itself, separate from the effects of BNNs. **For the complete algorithmic description and overview, please refer to Appendix B.**

## 4 EXPERIMENTS

This section presents practical applications of our `TokUR` for LLM reasoning. **For additional experimental results, please refer to Appendix E.**

**Datasets.** We run our main experiments on three mathematical reasoning benchmarks of varying difficulty levels: **GSM8K** (Cobbe et al., 2021) (grade-school arithmetic problems), **MATH500** (Lightman et al., 2023) (challenging high school/college mathematics competition problems), and 5,000-example subset of **DeepScaleR** (Luo et al., 2025) (high-difficulty problems from diverse sources). For these complex math problems, LLMs often need to perform multi-step reasoning (Wei et al., 2022b; Yao et al., 2023; Zhou et al., 2023) to reach the final answer. These tasks inherently involve long-form generation, therefore well-suited for evaluation of uncertainty estimation methods.

To assess the generalization of `TokUR` beyond mathematical reasoning, we further evaluate `TokUR` on five non-math long-form generation tasks, spanning **logical reasoning**, **code generation**, and **truthfulness evaluation**. For logical reasoning, we use three tasks from Reasoning Gym (Stojanovski et al., 2025): *Zebra Puzzles*, *Leg Counting*, and *Color-Cube Rotation*. For code generation, we evaluate on the **HumanEval** (Chen, 2021) benchmark, a widely adopted standard for functional code synthesis. For truthfulness, we use the **FactScore** (Min et al., 2023) dataset, which measures factual consistency by decomposing generated outputs into atomic facts; we follow prior work and use `GPT-5-mini` as both the fact annotator and judge.

**Models.** We evaluate our `TokUR` using models from two open-source LLM families: `Llama` (`3.2-1B-Instruct` and `3.1-8B-Instruct`) (Grattafiori et al., 2024) and `Qwen` (`2.5-3B-Instruct` and `2.5-7B-Instruct`) (Team et al., 2024). These models represent recent advances in open-source instruction tuning and provide a practical balance between capability and efficiency. Their differing **model scales and architectural families** further enable us to examine the consistency of uncertainty estimation across both model sizes and model types.

**Implementation of our `TokUR`.** We estimate token-level uncertainties by applying random perturbations as in Eqn. 17 to the query and key weight matrices ($\boldsymbol{W}^Q, \boldsymbol{W}^K$) (Vaswani et al., 2017) in all the attention layers of LLMs (Hu et al., 2022; Yang et al., 2023; Wang et al., 2024; Shi et al., 2024). For more details, please refer to Appendix D.1.

### 4.1 DO `TokUR`'S UNCERTAINTIES ACCURATELY REFLECT RESPONSE QUALITY?

This section assesses if our `TokUR`'s uncertainties reflect response quality in math reasoning tasks.

#### 4.1.1 `TokUR`'S UNCERTAINTIES AND QUESTION DIFFICULTY

**Experimental Setting.** To better understand the relationships among uncertainty estimates, question difficulty, and their ability to distinguish correct from incorrect responses, we sample a subset of

math questions from math-orz (Hu et al., 2025). A question's difficulty level is determined by the number of failed attempts out of 10 when using the `Qwen2.5-3B-Instruct` model. A difficulty level of 0 means the model solved the question every time, while a level of 10 indicates it failed on every attempt. We sample 500 questions per difficulty level, yielding a 5,500-question dataset. We then prompt `Llama-3.2-1B-Instruct` to solve each question with greedy decoding and apply `TokUR` to compute uncertainties for both correct and incorrect responses across difficulty levels. Fig. 1 summarizes the results.

**Results.** `TokUR`'s uncertainty estimates remain positively correlated with question difficulty: for all three types of uncertainty (AU, TU, and EU), incorrect responses consistently exhibit higher uncertainty than correct ones across difficulty levels. In terms of discriminative power, AUROC values are consistently above random (0.5), confirming that `TokUR` provides useful signals for distinguishing correct from incorrect reasoning. Yet, AUROC tends to decrease as difficulty increases, especially in the mid-to-high range (levels 7–9), showing that uncertainty estimates become less reliable at separating outcomes on challenging tasks. Interestingly, the AUROC score shows a slight increase at the highest difficulty level (10). This is likely a result of the imbalanced data distribution (mostly incorrect), where the model consistently produces high uncertainty, which causes a misleading high metric value.

### 4.1.2 TOKUR FOR INCORRECT REASONING PATH DETECTION

**Experimental Setting.** The preliminary study demonstrates that our `TokUR`'s uncertainty estimation can reflect the quality of generated responses, with lower uncertainty generally associated with better outputs. One important application of uncertainty estimation is hallucination detection in LLMs (Farquhar et al., 2024; Kossen et al., 2024; Ye et al., 2025). In this context, we treat uncertainty as a scoring function to identify hallucinated (incorrect) responses for long-form reasoning tasks. We adopt three metrics: Area Under the Receiver Operating Characteristic Curve (**AUROC**), Area Under the Precision-Recall Curve (**AUPRC**), and **Top-50% ACC (ACC\*)** (Farquhar et al., 2024; Ye et al., 2025; Hanley & McNeil, 1982; Boyd et al., 2013). AUROC and AUPRC measure the overall power of uncertainty scores in distinguishing correct from incorrect responses. In addition, we report Top-50% ACC, defined as the accuracy of the top 50% samples ranked by the corresponding score. This metric reflects the model's ability to prioritize higher-quality generations under a fixed budget. We repeat the experiments with three different random seeds to obtain the mean and standard deviation across runs.

**Baselines.** We systematically categorize our baselines into two distinct types: (i) those relying solely on the LLM's internal signals, including the most recent **Self-Certainty** (Kang et al., 2025), Deep Think With Confidence (**DeepConf**) (Fu et al., 2025), **LLM-Check** (Sriramanan et al., 2024), Degree Matrix ($U_{\text{Ecc}}$ and $U_{\text{Deg}}$) (Lin et al., 2023), and INternal States for hallucInation DEtection (**INSIDE**) (Chen et al., 2024a), as well as classic **P(True)** (Kadavath et al., 2022), Predictive Entropy (**PE**) (Malinin & Gales, 2021), Log-Likelihood (**LL**) (Murray & Chiang, 2018), and (ii) those leveraging external signals, such as an auxiliary Natural Language Inference model (He et al., 2020): Semantic Entropy (**SE**) (Kuhn et al., 2023), and Shifting Attention to Relevance (**SAR**) (Duan et al., 2024). Note that, apart from the five baselines with underlines (Kang et al., 2025; Fu et al., 2025; Kadavath et al., 2022; Malinin & Gales, 2021; Murray & Chiang, 2018), the others were originally designed for query-level de-hallucination in short-form QA tasks and are therefore not directly comparable to `TokUR`; we include them for completeness (see Appendix D.4 for details).

**Results.** As shown in Table 1, our proposed `TokUR` consistently outperforms baselines across AU-ROC, AUPRC, and ACC\*. For example, on `Llama-3.2-1B-Instruct`, `TokUR` (TU) achieves an AUROC of **80.64%** and an AUPRC of **56.79%** on MATH500, clearly surpassing all baselines. On the larger `Llama-3.1-8B-Instruct`, the improvements are also substantial: `TokUR` (EU) attains **82.86%** AUROC and **81.35%** AUPRC on MATH500, establishing new state-of-the-art performance. These results highlight an important insight: `TokUR` provides a reliable and scalable uncertainty estimation framework, achieving strong performance without relying on external signals. **For results of `Qwen` models, please see Appendix E.2.**

### 4.1.3 EXPERIMENTS BEYOND MATHEMATICAL REASONING

**Experimental Setting.** In this section, we evaluate the generalization ability of `TokUR` on domains beyond mathematical reasoning. For verifiable tasks such as Logical Reasoning and Code Generation,

Table 1: **Performance of Uncertainty Estimation Methods for Incorrect Reasoning Path Detection.** AUROC, AUPRC, and ACC* are all reported as percentage (%). **"ISO?"** indicates whether the method utilizes **I**nternal **S**ignal **O**nly for uncertainty estimation. We include the accuracy of CoT (i.e., greedy decoding with Chain-of-Thought prompting) in the first row for reference. **Boldface** and underlining denote the best and the second-best performance, respectively.

| Method | ISO? | MATH500 | | | GSM8K | | | DeepScaleR | | |
|---|---|---|---|---|---|---|---|---|---|---|
| | | AUROC | AUPRC | ACC* | AUROC | AUPRC | ACC* | AUROC | AUPRC | ACC* |
| *Llama-3.2-1B-Instruct* | | | | | | | | | | |
| CoT (Lower-Bound) | - | - | - | 25.60±0.00 | - | - | 44.43±0.00 | - | - | 14.25±0.00 |
| SE | ✗ | 47.29±3.81 | 25.71±2.33 | 24.13±4.42 | 50.64±4.44 | 45.09±0.72 | 42.62±0.16 | 46.30±0.21 | 12.94±0.23 | 12.58±0.49 |
| SAR | ✗ | 44.57±2.04 | 24.03±2.53 | 21.07±1.62 | 50.28±0.97 | 43.24±0.89 | 43.95±0.77 | 43.14±1.42 | 12.34±0.35 | 11.14±0.47 |
| $U_{Ecc}$ | ✗ | 48.75±1.05 | 25.79±1.83 | 25.20±0.33 | 49.05±0.46 | 60.02±0.44 | 59.62±0.22 | 48.68±0.24 | 13.77±0.29 | 14.23±0.45 |
| $U_{Deg}$ | ✗ | 60.57±2.31 | 36.32±2.59 | 30.93±0.94 | 66.60±0.36 | 75.72±0.36 | 71.99±0.39 | 56.88±0.54 | 18.04±0.63 | 16.50±0.39 |
| P(True) | ✓ | 54.38±1.20 | 26.39±1.26 | 27.60±1.18 | 56.64±0.04 | 48.22±0.03 | 48.92±0.00 | 59.58±0.43 | 17.48±0.25 | 17.52±0.50 |
| LLM-Check | ✓ | 56.41±0.96 | 27.01±1.22 | 31.33±1.29 | 71.01±0.02 | 61.29±0.08 | 59.54±0.00 | 55.76±0.48 | 14.55±0.26 | 17.30±0.51 |
| INSIDE | ✓ | 55.71±4.69 | 28.82±4.05 | 29.20±4.33 | 53.66±0.92 | 46.03±0.23 | 45.79±1.25 | 54.73±0.82 | 15.50±0.48 | 16.30±0.35 |
| PE | ✓ | 57.08±0.89 | 26.88±1.05 | 31.33±0.82 | 71.21±0.03 | 61.61±0.08 | 59.85±0.00 | 56.09±0.46 | 14.74±0.23 | 17.33±0.92 |
| LL | ✓ | 55.41±0.54 | 25.88±0.87 | 29.87±0.82 | 69.01±0.03 | 58.51±0.09 | 57.38±0.00 | 53.84±0.47 | 13.93±0.23 | 16.83±0.48 |
| Self-Certainty | ✓ | 71.17±0.30 | 48.37±0.50 | 38.13±0.61 | 73.41±0.00 | 68.38±0.00 | 61.38±0.00 | 71.93±0.04 | 33.81±0.08 | 21.76±0.04 |
| DeepConf | ✓ | 71.77±0.12 | 46.00±0.42 | 39.87±0.46 | **75.70**±0.00 | 69.72±0.00 | **62.77**±0.00 | 71.65±0.04 | 29.99±0.05 | 22.00±0.04 |
| TokUR (TU, Ours) | ✓ | **80.64**±0.29 | **56.79**±0.74 | **44.67**±0.46 | 75.07±0.05 | **70.29**±0.07 | 62.31±0.00 | **83.55**±0.02 | **47.56**±0.04 | **25.71**±0.02 |
| TokUR (AU, Ours) | ✓ | 80.61±0.27 | 56.73±0.75 | **44.67**±0.46 | 75.03±0.06 | 70.22±0.05 | 62.21±0.18 | 83.52±0.02 | 47.48±0.05 | **25.71**±0.02 |
| TokUR (EU, Ours) | ✓ | 79.74±0.21 | 56.64±0.41 | 44.13±0.83 | 71.79±0.80 | 66.40±1.02 | 59.74±1.00 | 82.87±0.32 | 46.76±0.38 | 25.52±0.11 |
| *Llama-3.1-8B-Instruct* | | | | | | | | | | |
| CoT (Lower-Bound) | - | - | - | 48.60±0.00 | - | - | 85.69±0.00 | - | - | 24.86±0.00 |
| SE | ✗ | 62.93±0.90 | 55.21±1.04 | 55.73±0.83 | 55.61±3.36 | 87.16±1.14 | 86.77±1.01 | 67.68±0.94 | 35.18±1.00 | 35.55±0.37 |
| SAR | ✗ | 69.42±2.19 | 63.74±3.03 | 59.20±1.06 | 60.16±2.22 | 89.24±0.74 | 87.99±0.81 | 73.01±0.28 | 42.89±0.65 | 37.51±0.12 |
| $U_{Ecc}$ | ✗ | 50.23±2.23 | 49.48±2.44 | 49.60±2.04 | 47.47±2.15 | 84.69±0.89 | 84.87±1.17 | 50.16±0.66 | 25.08±0.18 | 25.48±0.53 |
| $U_{Deg}$ | ✗ | 58.62±0.36 | 57.69±0.90 | 53.47±1.64 | 67.22±1.06 | 92.24±0.53 | 92.62±0.88 | 59.14±0.37 | 32.64±0.43 | 29.75±0.36 |
| P(True) | ✓ | 33.41±0.25 | 36.05±0.55 | 35.33±0.19 | 41.94±0.01 | 82.19±0.00 | 82.77±0.00 | 33.64±0.20 | 18.06±0.06 | 16.23±0.02 |
| LLM-Check | ✓ | 57.41±0.44 | 49.69±1.07 | 52.80±1.38 | 73.98±0.01 | 93.37±0.01 | 93.23±0.00 | 55.42±0.27 | 26.46±0.19 | 28.37±0.40 |
| INSIDE | ✓ | 62.94±1.72 | 55.06±3.19 | 57.33±1.01 | 58.86±2.11 | 87.44±0.94 | 88.21±0.90 | 67.05±0.49 | 33.83±0.42 | 34.13±0.10 |
| PE | ✓ | 57.98±0.49 | 49.72±0.84 | 53.07±0.94 | 74.03±0.01 | 93.37±0.00 | 93.23±0.00 | 55.90±0.23 | 26.80±0.16 | 28.65±0.22 |
| LL | ✓ | 55.36±0.49 | 47.24±0.90 | 51.07±0.94 | 72.21±0.02 | 92.64±0.00 | 92.46±0.00 | 52.82±0.32 | 24.48±0.13 | 26.85±0.19 |
| Self-Certainty | ✓ | 76.41±0.61 | 76.22±0.87 | 69.07±0.83 | 80.60±0.11 | 95.65±0.03 | 96.26±0.09 | 76.72±0.09 | 56.15±0.30 | 39.03±0.23 |
| DeepConf | ✓ | 71.86±0.70 | 69.57±0.94 | 66.27±1.15 | **83.30**±0.07 | **96.23**±0.02 | **96.56**±0.09 | 73.05±0.00 | 48.76±0.10 | 37.48±0.14 |
| TokUR (TU, Ours) | ✓ | 82.47±0.47 | 79.62±0.33 | **74.00**±0.69 | 81.01±0.04 | 95.53±0.05 | 95.54±0.00 | **85.33**±0.07 | 65.25±0.01 | **43.91**±0.09 |
| TokUR (AU, Ours) | ✓ | 82.43±0.48 | 79.56±0.35 | **74.00**±0.69 | 80.97±0.02 | 95.52±0.03 | 95.49±0.09 | 85.31±0.07 | 65.20±0.02 | 43.89±0.08 |
| TokUR (EU, Ours) | ✓ | **82.86**±0.42 | **81.35**±0.66 | 72.40±1.20 | 78.31±1.58 | 94.91±0.59 | 94.67±0.77 | 84.92±0.28 | **65.57**±0.43 | 43.89±0.27 |

Table 2: **Performance of Uncertainty Estimation Methods on Logical Reasoning Tasks.** We report AUROC, AUPRC, and ACC*, all reported as percentage (%), for three logical reasoning tasks from Reasoning Gym. **Boldface** and underlining denote the best and second-best performance.

| Method | Zebra Puzzles | | | Leg Counting | | | Color Cube | | |
|---|---|---|---|---|---|---|---|---|---|
| | AUROC | AUPRC | ACC* | AUROC | AUPRC | ACC* | AUROC | AUPRC | ACC* |
| CoT (Lower-Bound) | - | - | 33.67±0.00 | - | - | 35.00±0.00 | - | - | 26.00±0.00 |
| PE | 44.70±0.00 | 21.76±0.00 | 24.00±0.00 | 38.87±0.25 | 27.44±0.05 | 26.00±0.00 | 46.73±0.00 | 28.75±0.00 | 22.00±0.00 |
| LL | 42.15±0.00 | 21.02±0.00 | 22.00±0.00 | 37.92±0.43 | 27.10±0.13 | 32.67±1.15 | 46.36±0.00 | 26.01±0.00 | 24.00±0.00 |
| Self-Certainty | 47.77±0.00 | 22.95±0.00 | 30.00±0.00 | 50.46±0.30 | 32.88±0.37 | 34.00±0.00 | 50.62±0.00 | 30.80±0.00 | 28.00±0.00 |
| DeepConf | 42.41±0.00 | 21.07±0.00 | 26.00±0.00 | 42.48±0.71 | 28.96±0.32 | 30.00±0.00 | 49.01±0.00 | 24.41±0.00 | 24.00±0.00 |
| TokUR (TU, Ours) | 71.38±0.29 | 41.42±0.38 | **39.33**±1.15 | 69.58±2.05 | 48.16±0.99 | 48.67±1.15 | 58.33±2.86 | 31.14±1.56 | 31.33±1.15 |
| TokUR (AU, Ours) | **71.66**±0.35 | **41.71**±0.44 | **39.33**±1.15 | 69.51±1.82 | 47.69±0.70 | 48.67±1.15 | 56.57±2.88 | 30.02±1.36 | 28.00±2.00 |
| TokUR (EU, Ours) | 71.28±0.38 | 41.20±0.76 | 37.33±2.31 | **69.66**±2.51 | **51.27**±2.07 | **50.67**±2.31 | **60.08**±1.98 | **33.71**±1.01 | **32.00**±2.00 |

we report **AUROC**, **AUPRC**, and **Top-50% ACC (ACC*)**, following the protocol described in Sec. 4.1.2. For the Truthfulness task, we report **Top-50% Precision (Precision*)** based on uncertainty-ranked responses. Due to the increased difficulty of these tasks, all evaluations are conducted using `Qwen-2.5-3B-Instruct`.

**Results.** The results are summarized in Table 2 (logical reasoning) and Table 3 (code generation and truthfulness). Across all non-mathematical tasks, `TokUR` consistently ranks as the **best or second-best method**, demonstrating that it generalizes robustly to diverse long-form generation settings while maintaining strong performance beyond the mathematical reasoning domain.

## 4.2 CAN `TokUR`'S UNCERTAINTIES IMPROVE GENERATION QUALITY?

In this section, we explore the direct application of `TokUR` to reasoning tasks to enhance generation quality. Following previous works (Fu et al., 2025), we apply `TokUR` to measure the confidence of reasoning traces generated from a question and aggregate them via voting to obtain a final solution.

Table 3: **Performance of Uncertainty Estimation Methods on Code Generation and Truthfulness Tasks.** HumanEval results report AUROC, AUPRC, and ACC*, all reported as percentage (%); FactScore reports Precision*(%). **Boldface** marks best results.

| Method | HumanEval | | | FactScore |
|---|---|---|---|---|
| | AUROC | AUPRC | ACC* | Precision* |
| CoT (Lower-Bound) | - | - | $74.39_{\pm0.00}$ | $52.43_{\pm0.00}$ |
| PE | $60.52_{\pm0.00}$ | $81.45_{\pm0.00}$ | $78.05_{\pm0.00}$ | $41.86_{\pm0.02}$ |
| LL | $56.42_{\pm0.00}$ | $79.48_{\pm0.00}$ | $75.61_{\pm0.00}$ | $45.03_{\pm0.38}$ |
| Self-Certainty | $61.48_{\pm0.00}$ | $80.27_{\pm0.00}$ | $\underline{80.49}_{\pm0.00}$ | $40.10_{\pm0.38}$ |
| DeepConf | $\mathbf{63.60}_{\pm0.00}$ | $\mathbf{82.75}_{\pm0.00}$ | $\underline{80.49}_{\pm0.00}$ | $41.96_{\pm0.49}$ |
| TokUR (TU, Ours) | $\underline{63.15}_{\pm0.82}$ | $81.59_{\pm0.41}$ | $\mathbf{80.90}_{\pm0.70}$ | $\underline{60.35}_{\pm0.74}$ |
| TokUR (AU, Ours) | $60.50_{\pm0.80}$ | $\underline{81.75}_{\pm0.41}$ | $\mathbf{80.90}_{\pm1.41}$ | $\mathbf{61.53}_{\pm0.58}$ |
| TokUR (EU, Ours) | $60.64_{\pm0.77}$ | $80.75_{\pm0.48}$ | $\underline{80.49}_{\pm0.00}$ | $45.26_{\pm0.54}$ |

Table 4: **Performance of Uncertainty Estimation Methods for Test-Time Scaling. Boldface** and underlining denote the best and the second-best performance, respectively.

| Dataset | Score | Method | Llama-3.2-1B-Instruct | | | Llama-3.1-8B-Instruct | | |
|---|---|---|---|---|---|---|---|---|
| | | | N=16 | N=64 | N=256 | N=16 | N=64 | N=256 |
| **GSM8K** (Pass@1: 44.43 / 85.69) | LL | Maj@N | $47.10_{\pm0.85}$ | $54.11_{\pm0.52}$ | $58.89_{\pm0.36}$ | $86.74_{\pm0.62}$ | $90.48_{\pm0.48}$ | $91.01_{\pm0.28}$ |
| | | WBoN | $47.10_{\pm0.85}$ | $54.15_{\pm0.55}$ | $58.92_{\pm0.37}$ | $86.74_{\pm0.62}$ | $90.48_{\pm0.49}$ | $91.00_{\pm0.29}$ |
| | Self-Certainty | Maj@N | $45.02_{\pm0.92}$ | $52.61_{\pm0.72}$ | $57.18_{\pm0.53}$ | $80.02_{\pm0.70}$ | $87.25_{\pm0.49}$ | $90.05_{\pm0.40}$ |
| | | WBoN | $45.02_{\pm0.92}$ | $52.65_{\pm0.70}$ | $57.22_{\pm0.54}$ | $80.02_{\pm0.70}$ | $87.25_{\pm0.50}$ | $90.05_{\pm0.41}$ |
| | DeepConf | Maj@N | $46.72_{\pm0.89}$ | $53.50_{\pm0.66}$ | $58.05_{\pm0.44}$ | $86.24_{\pm0.66}$ | $90.34_{\pm0.46}$ | $90.92_{\pm0.28}$ |
| | | WBoN | $46.72_{\pm0.89}$ | $53.47_{\pm0.65}$ | $58.10_{\pm0.45}$ | $86.24_{\pm0.66}$ | $90.32_{\pm0.46}$ | $91.02_{\pm0.00}$ |
| | TokUR (TU, Ours) | Maj@N | $50.29_{\pm1.03}$ | $57.18_{\pm0.46}$ | $60.68_{\pm0.49}$ | $87.68_{\pm0.57}$ | $90.67_{\pm0.45}$ | $90.96_{\pm0.36}$ |
| | | WBoN | $\underline{50.29}_{\pm1.03}$ | $\mathbf{57.22}_{\pm0.45}$ | $\underline{60.71}_{\pm0.49}$ | $\underline{87.68}_{\pm0.57}$ | $90.65_{\pm0.46}$ | $90.98_{\pm0.37}$ |
| | TokUR (AU, Ours) | Maj@N | $50.20_{\pm0.98}$ | $\underline{57.21}_{\pm0.46}$ | $60.70_{\pm0.41}$ | $87.42_{\pm0.66}$ | $90.60_{\pm0.44}$ | $90.99_{\pm0.32}$ |
| | | WBoN | $50.20_{\pm0.98}$ | $57.19_{\pm0.44}$ | $\mathbf{60.78}_{\pm0.42}$ | $87.42_{\pm0.66}$ | $90.57_{\pm0.43}$ | $91.02_{\pm0.00}$ |
| | TokUR (EU, Ours) | Maj@N | $\mathbf{50.38}_{\pm0.92}$ | $56.92_{\pm0.60}$ | $59.88_{\pm0.52}$ | $\mathbf{88.06}_{\pm0.57}$ | $\mathbf{90.69}_{\pm0.47}$ | $\underline{91.07}_{\pm0.33}$ |
| | | WBoN | $\mathbf{50.38}_{\pm0.92}$ | $56.89_{\pm0.54}$ | $59.91_{\pm0.58}$ | $\mathbf{88.06}_{\pm0.57}$ | $\underline{90.67}_{\pm0.48}$ | $\mathbf{91.09}_{\pm0.36}$ |
| **MATH500** (Pass@1: 25.60 / 48.60) | LL | Maj@N | $26.42_{\pm0.84}$ | $33.28_{\pm0.97}$ | $38.56_{\pm0.75}$ | $50.92_{\pm1.77}$ | $59.36_{\pm0.74}$ | $64.10_{\pm0.61}$ |
| | | WBoN | $26.42_{\pm0.84}$ | $33.30_{\pm1.10}$ | $38.58_{\pm0.73}$ | $50.92_{\pm1.77}$ | $59.46_{\pm0.78}$ | $64.02_{\pm0.71}$ |
| | Self-Certainty | Maj@N | $20.14_{\pm1.14}$ | $29.12_{\pm1.11}$ | $36.68_{\pm0.83}$ | $44.00_{\pm1.82}$ | $55.56_{\pm1.08}$ | $62.66_{\pm0.75}$ |
| | | WBoN | $20.14_{\pm1.14}$ | $29.16_{\pm0.99}$ | $36.80_{\pm0.80}$ | $44.00_{\pm1.82}$ | $55.58_{\pm1.06}$ | $62.52_{\pm0.53}$ |
| | DeepConf | Maj@N | $25.68_{\pm1.38}$ | $33.30_{\pm1.10}$ | $38.52_{\pm0.43}$ | $49.88_{\pm1.29}$ | $59.74_{\pm1.17}$ | $64.30_{\pm0.63}$ |
| | | WBoN | $25.68_{\pm1.38}$ | $32.44_{\pm1.20}$ | $37.08_{\pm0.78}$ | $49.88_{\pm1.29}$ | $58.22_{\pm1.00}$ | $63.14_{\pm0.55}$ |
| | TokUR (TU, Ours) | Maj@N | $27.06_{\pm0.94}$ | $\underline{33.76}_{\pm0.84}$ | $39.18_{\pm0.70}$ | $51.26_{\pm1.36}$ | $59.44_{\pm1.31}$ | $63.86_{\pm0.44}$ |
| | | WBoN | $\underline{27.06}_{\pm0.94}$ | $33.60_{\pm0.82}$ | $39.20_{\pm0.65}$ | $\underline{51.26}_{\pm1.36}$ | $59.44_{\pm1.30}$ | $63.84_{\pm0.51}$ |
| | TokUR (AU, Ours) | Maj@N | $27.06_{\pm0.91}$ | $33.64_{\pm0.76}$ | $39.12_{\pm0.72}$ | $51.16_{\pm1.45}$ | $59.42_{\pm1.16}$ | $64.00_{\pm0.44}$ |
| | | WBoN | $\underline{27.06}_{\pm0.91}$ | $33.48_{\pm0.73}$ | $39.10_{\pm0.69}$ | $51.16_{\pm1.45}$ | $59.44_{\pm1.19}$ | $63.92_{\pm0.47}$ |
| | TokUR (EU, Ours) | Maj@N | $\mathbf{28.28}_{\pm1.32}$ | $\mathbf{35.44}_{\pm0.79}$ | $\mathbf{39.44}_{\pm0.88}$ | $\mathbf{52.40}_{\pm1.39}$ | $\mathbf{60.90}_{\pm0.93}$ | $\mathbf{65.32}_{\pm0.80}$ |
| | | WBoN | $\mathbf{28.28}_{\pm1.32}$ | $\mathbf{35.44}_{\pm0.78}$ | $\underline{39.38}_{\pm0.87}$ | $\mathbf{52.40}_{\pm1.39}$ | $\mathbf{61.04}_{\pm0.88}$ | $\mathbf{65.48}_{\pm0.75}$ |

In addition, we investigate the possibility of utilizing TokUR in an *online* manner to dynamically guide the generation process itself. Further details of online method are provided in E.4.

**Baselines.** We adopt Log-Likelihood (**LL**) as a baseline, given its widespread use as a proxy for generation quality (Manakul et al., 2023; Rafailov et al., 2023; Chen et al., 2024b). In addition, we compare against **Self-Certainty** (Kang et al., 2025) and **DeepConf** (Fu et al., 2025), two recent uncertainty-driven approaches for test-time scaling. As our study emphasizes model self-awareness of the boundaries of its knowledge, we do not include baselines that rely on external reward models (Guan et al., 2025; Puri et al., 2025; Beeching et al.; Uesato et al., 2022; Lightman et al., 2023).

**Response Aggregation with Uncertainties.** We first rank all $N$ candidate responses using one of the scoring methods (**LL**, **Self-Certainty**, **DeepConf**, or TokUR) and retain the top-$P\%$ candidates. We then employ two common aggregation strategies: *Weighted Best-of-N* (**WBoN**) and *Majority Voting* (**Maj@N**) (Brown et al., 2024). WBoN performs weighted voting by assigning weights to the retained candidates according to their scores, whereas Maj@N simply selects the most frequent response among them, regardless of scoring.

**Experimental Setting**. We randomly sample 512 responses for each question in MATH500 and GSM8K with a decoding temperature of $\tau = 0.8$. For each $N$, we first retain the **top-10%** of samples ranked by their scores. From this subset, the final prediction is determined using either Maj@N or WBoN. Each experiment is repeated 10 times (sample w/o replacement using offline records).

**Results**. As shown in Table 4, accuracy consistently improves with larger $N$ across both GSM8K and MATH500. Our `TokUR`-based selection methods achieve clear gains over all baselines, particularly in the low-sample regime ($N$=16), where they deliver up to 3–4 points of improvement. Notably, `TokUR` (EU) attains the best overall performance on both datasets, with strong advantages in the challenging MATH500 benchmark. In addition, results for Maj@N and WBoN are similar, indicating that both aggregation strategies are similarly effective once the top candidates are identified.

## 5 RELATED WORK

**Uncertainty Estimation of LLMs.** Uncertainty estimation in LLMs is gaining traction for improving model calibration in data-scarce adaptation tasks and for reducing hallucinations in text generation (Liu et al., 2025; Vashurin et al., 2025). One prominent approach is Bayesian Adaptation, which combines Bayesian inference with low-rank adaptation (LoRA) (Hu et al., 2022) to approximate weight posterior distributions efficiently, avoiding the high computational cost of full Bayesian modeling (Yang et al., 2023; Wang et al., 2024; Shi et al., 2024). To estimate uncertainty in generation, two main lines of work have emerged. The first focuses on *verbalized uncertainty,* where models are prompted to express confidence in natural language (Lin et al., 2022; Kadavath et al., 2022; Tian et al., 2023; Kapoor et al., 2024). The second line includes *logits-based methods,* which estimate uncertainty directly from the model's output distributions (Van Der Poel et al., 2022; Ren et al., 2023; Duan et al., 2024; Darrin et al., 2023). In parallel, other approaches aim to refine these estimation strategies. For instance, Malinin & Gales (2021) investigates techniques for estimating epistemic uncertainty in structured prediction tasks, while semantic entropy (Kuhn et al., 2023) captures uncertainty by leveraging invariance in meaning across paraphrases. More recently, Zhang & Zhang (2025) introduces a method that leverages the reasoning capabilities of LLMs to enhance uncertainty quantification, using chain-of-thought prompting to better reflect model confidence in multi-step tasks. These works complement verbalized and logits-based methods by offering orthogonal perspectives on how uncertainty can be interpreted and measured. Beyond specific estimation methods, recent studies reassess broader principles of LLM uncertainty: Kirchhof et al. (2025) argues that uncertainty quantification for LLM agents requires rethinking foundational assumptions, while Devic et al. (2025) highlights the limitations of existing approaches and calls for more human-centered uncertainty interpretation.

**Uncertainty for Improving LLM Generation.** Uncertainty estimation for improving LLM generation, while not entirely novel, has been predominantly limited to approaches based on log-probability or its variants, Self-Certainty (Kang et al., 2025) estimates confidence via KL divergence from a uniform distribution, DeepConf (Fu et al., 2025) aggregates top-$K$ log-probabilities as scores. Beam search (Lowerre, 1976; Sutskever et al., 2014; Freitag & Al-Onaizan, 2017; Xie et al., 2023) selects higher-confidence sequences by retaining candidates with the largest cumulative log-probability. UAG (Yin et al., 2024) leverages abrupt log-probability changes to select appropriate demonstrations for in-context learning (Brown et al., 2020). UnCert-CoT (Zhu et al., 2025) alternates between greedy and Chain-of-Thought decoding based on log-probability scores. Complementing these, Zhou et al. (2025) provides theoretical insights into how internal probabilities relate to self-consistency and reasoning errors in LLMs. Our work differs fundamentally by estimating token-level uncertainties with rigorous theoretical foundations, representing a significant step toward extending Bayesian LLMs to long-form generation scenarios.

**Limitations.** Our work is subject to several limitations. First, although compatible with efficient deployment frameworks such as vLLM (Kwon et al., 2023), repeated weight perturbation sampling during inference still poses efficiency challenges for real-time use. Second, our token-level uncertainty aggregation may miss higher-level semantic or logical inconsistencies across multiple tokens or reasoning steps, limiting its utility in complex generation tasks. Finally, the problem of high-variance estimation in our `TokUR` remains unresolved, constraining reliability in real-world scenarios.

## ACKNOWLEDGEMENT

We thank all reviewers, AC, and SAC for their valuable comments. HW is supported by Amazon Faculty Research Award, Microsoft AI & Society Fellowship, NSF CAREER Award IIS-2340125, NIH grant R01CA297832, and NSF grant IIS-2127918.

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

# APPENDIX

In Appendix A, we describe the role of large language models (LLMs) in our work. In Appendix B, we present the full algorithmic description of our method with low-rank weight perturbation. In Appendix C, we provide detailed proofs for all propositions presented in the main paper. In Appendix D, we provide our implementation details of the experiments, including:

- **implementation of our `TokUR`** (Appendix D.1),
- **dataset details** (Appendix D.2),
- **prompt templates** used in LLM reasoning (Appendix D.3),
- **baseline details** (Appendix D.4),
- and **evaluation metrics** (Appendix D.5).

Finally, in Appendix E, we present additional empirical results, including:

- **preliminary study** on the uncertainty distributions produced by TokUR (Appendix E.1),
- additional results of **Qwen** models (Appendix E.2),
- **detailed numerical results** of the test-time scaling (Appendix E.3),
- **online test-time scaling of `TokUR`** (Appendix E.4),
- **an ablation study** on different components of our token-level uncertainties (Appendix E.5),
- and **a case study** of our token-level uncertainties (Appendix E.6).

## A    LLM USAGE DISCLOSURE

We used large language models (LLMs) solely to assist with polishing the writing of this paper, including improving grammar, clarity, and readability. The LLMs did not contribute to research ideation, experimental design, analysis, or the generation of scientific content. All technical contributions, claims, and conclusions are the authors' own.

## B    ALGORITHM DETAILS

---

**Algorithm 1** Low-Rank Weight Perturbation as Approximation of Weight Posterior.

---
1: **Input**
2:     The base model policy $p(\mathbf{y}|\mathbf{x})$;
3:     The set of weight matrices to be Bayesianized $\{\boldsymbol{W}_0^k\}_{k=1}^N$;
4:     rank of noise matrix $r'$;
5:     The perturbation strength $\sigma_q$.
6: **for** $i = 1$ to $N$ **do**
7:        $\boldsymbol{U}, \mathrm{diag}(\boldsymbol{d}), \boldsymbol{V}^\top \leftarrow \mathrm{SVD}(\boldsymbol{W}_0^k)$.                                    ▷ Eqn. 16
8:        $\boldsymbol{U}' \leftarrow$ the first $r'$ columns of matrix $\boldsymbol{U}$.
9:        Sample noise matrix $\boldsymbol{\epsilon} \in \mathbb{R}^{n \times r'}$: $\epsilon_{ij} \sim N(0, \sigma_q)$.
10:       Perturb the weight matrix: $\boldsymbol{W}^k \leftarrow \boldsymbol{W}_0^k + \boldsymbol{U}'\boldsymbol{\epsilon}^\top$.                    ▷ Eqn. 17
11:       Get the weight posterior: $q(\mathrm{vec}(\boldsymbol{W}^k)|\sigma_q)$.                                ▷ Eqn. 18
12: **end for**
13: **Output:** The overall approximate posterior: $q(\boldsymbol{\theta}|\sigma_q) \leftarrow \prod_k q(\mathrm{vec}(\boldsymbol{W}^k)|\sigma_q)$

---

## C    PROOF OF PROPOSITIONS

**Lemma C.1** (**Definition of Conditional Entropy** (Cover, 1999)). *Give* $(\mathbf{y}, \mathbf{x}) \sim p(\mathbf{y}, \mathbf{x})$, *the conditional entropy* $\mathcal{H}(\mathbf{y}|\mathbf{x})$ *is defined as*

$$\begin{aligned}
\mathcal{H}(\mathbf{y}|\mathbf{x}) &= \sum_{\boldsymbol{x} \in \mathcal{X}} p(\boldsymbol{x}) \mathcal{H}(\mathbf{y}|\boldsymbol{x}) \\
&= \mathbb{E}_{\boldsymbol{x} \sim p(\mathbf{x})}[\mathcal{H}(\mathbf{y}|\boldsymbol{x})].
\end{aligned} \tag{20}$$

---

**Algorithm 2** Particle Filtering for Inference-Time Scaling (Puri et al., 2025)

---

1: **Input**
2:    The number of particles $N$;
3:    A reward model $\widehat{r}$;
4:    A LLM $p_M$ and a prompt $c$.
5: Initialize $N$ particles $\{x_1^i \sim p_M(\cdot|c)\}_{i=1}^N$.
6: $t \leftarrow 1$.
7: **while** not all particles stop **do**
8:    Update rewards $\mathbf{w} = \{\widehat{r}(x_{1:t}^{(1)}), \widehat{r}(x_{1:t}^{(2)}), \ldots, \widehat{r}(x_{1:t}^{(N)})\}$.
9:    Compute softmax distribution $\boldsymbol{\theta} = \text{softmax}(\mathbf{w})$.
10:    Sample indices $\{j_t^{(i)}\}_{i=1}^N \sim \mathbb{P}_t(j = i) = \boldsymbol{\theta}_i$.
11:    Update the set of particles as $\{x_{1:t}^{(j_t^{(I)})}\}_{i=1}^N$.
12:    Transition $\{x_{t+1}^i \sim p_M(\cdot|c, x_{1:t}^{(i)})\}_{i=1}^N$.
13:    $t \leftarrow t + 1$.
14: **end while**
15: **Output:** The set of particles in the end.

---

**Lemma C.2** (**Chain rule of Conditional Entropy** (Cover, 1999)). *Let $\boldsymbol{X}$ and $\boldsymbol{Y}$ be two random variables, then the conditional entropy of the joint distribution $\mathcal{H}(\boldsymbol{X}, \boldsymbol{Y})$ can be decomposed as:*

$$\mathcal{H}(\boldsymbol{X}, \boldsymbol{Y}) = \mathcal{H}(\boldsymbol{X}) + \mathcal{H}(\boldsymbol{Y}|\boldsymbol{X}) \tag{21}$$

Lemma C.1 (Cover, 1999) reveals the relationship between conditional entropy $\mathcal{H}(\mathbf{y}|\mathbf{x})$ and the entropy derived from conditional probability distributions. Lemma C.2 lays the foundation for estimating the uncertainties of sequences. The two lemmas together give us the following proposition.

**Proposition C.1** (**Decomposition of Query-Level Uncertainty, Eqn. 4**). *Suppose that we have an input sequence $\boldsymbol{x}$ and a model policy $p(\mathbf{y}|\mathrm{x})$. The sequence-level uncertainty $\mathcal{U}(\mathbf{y}|\boldsymbol{x})$ can be decomposed token-by-token as:*

$$\mathcal{U}(\mathbf{y}|\boldsymbol{x}) = \sum_{t=1}^T \mathcal{U}(\mathbf{y}_t|\mathbf{y}_{<t}, \boldsymbol{x}), \tag{22}$$

*where $\mathcal{U}(\mathbf{y}_t|\boldsymbol{y}_{<t}, \boldsymbol{x})$ is token-level uncertainty metric as defined in Eqn. 10 ~ Eqn. 12.*

*Proof.* For Aleatoric Uncertainty (AU) and Total Uncertainty (TU) defined in Eqn. 10 and Eqn. 11, both are expressed in terms of entropy. Therefore, the decomposition of sequence-level uncertainty can be directly derived using the chain rule stated in the Lemma C.2.

For Epistemic Uncertainty (EU), also called *mutual information* defined in Eqn. 12, we proceed with the following derivation:

$$\mathcal{H}(p(\mathbf{y}|\boldsymbol{x})) = \mathcal{H}\Big(\mathbb{E}_{p(\boldsymbol{\theta}|\mathcal{D})}[p(\mathbf{y}_1|\boldsymbol{x}; \boldsymbol{\theta})] \cdot \cdots \cdot \mathbb{E}_{p(\boldsymbol{\theta}|\mathcal{D})}[p(\mathbf{y}_T|\mathbf{y}_{<T}, \boldsymbol{x}; \boldsymbol{\theta})]\Big) \tag{23}$$

$$= \sum_t^T \mathcal{H}(\mathbb{E}_{p(\boldsymbol{\theta}|\mathcal{D})}[p(\mathbf{y}_t|\mathbf{y}_{<t}, \boldsymbol{x}; \boldsymbol{\theta})]) \tag{24}$$

$$= \sum_t^T \mathcal{I}(\mathbf{y}_t; \boldsymbol{\theta}|\mathbf{y}_{<t}, \boldsymbol{x}) + \sum_t^T \mathbb{E}_{p(\boldsymbol{\theta}|\mathcal{D})}[\mathcal{H}(p(\mathbf{y}_t|\mathbf{y}_{<t}, \boldsymbol{x}; \boldsymbol{\theta}))] \tag{25}$$

$$= \sum_t^T \mathcal{I}(\mathbf{y}_t; \boldsymbol{\theta}|\mathbf{y}_{<t}, \boldsymbol{x}) + \mathbb{E}_{p(\boldsymbol{\theta}|\mathcal{D})}\mathcal{H}(p(\mathbf{y}|\boldsymbol{x}; \boldsymbol{\theta})) \tag{26}$$

Finally, based on the definition of *mutual information*, we obtain:

$$\mathcal{I}(\mathbf{y}; \boldsymbol{\theta}|\boldsymbol{x}) = \mathcal{H}(p(\mathbf{y}|\boldsymbol{x})) - \mathbb{E}_{p(\boldsymbol{\theta}|\mathcal{D})}\mathcal{H}(p(\mathbf{y}|\boldsymbol{x}; \boldsymbol{\theta}))$$

$$= \sum_t^T \mathcal{I}(\mathbf{y}_t; \boldsymbol{\theta}|\mathbf{y}_{<t}, \boldsymbol{x}) \tag{27}$$

$\square$

**Proposition 3.1** (**Response-Level Uncertainty as an Unbiased Estimator of Query-Level Uncertainty**). *Given an input query $\boldsymbol{x}$, let $\boldsymbol{y} \sim p(\mathbf{y}|\boldsymbol{x})$ be a generated sample of length $T$. Then the response-level uncertainty $\widetilde{\mathcal{U}}$ (Definition 3.1) is an **unbiased estimator** of the query-level uncertainty $\mathcal{U}$ (Definition 2.1), i.e.,*

$$\mathbb{E}_{\boldsymbol{y} \sim p(\mathbf{y}|\boldsymbol{x})}[\widetilde{\mathcal{U}}(\boldsymbol{y}|\boldsymbol{x})] = \mathcal{U}(\mathbf{y}|\boldsymbol{x}). \tag{28}$$

*Proof.* Based on Lemma C.1, for the token-level uncertainty $\mathcal{U}(\mathbf{y}_t|\boldsymbol{y}_{<t}, \boldsymbol{x})$ defined in Eqn. 10~Eqn. 12, we have

$$\mathbb{E}_{\boldsymbol{y}_{<t} \sim p(\cdot|\boldsymbol{x})}[\mathcal{U}(\mathbf{y}_t|\boldsymbol{y}_{<t}, \boldsymbol{x})] = \sum_{\boldsymbol{y}_{<t} \in \mathcal{Y}} p(\boldsymbol{y}_{<t}|\boldsymbol{x})\mathcal{U}(\mathbf{y}_t|\boldsymbol{y}_{<t}, \boldsymbol{x}) \tag{29}$$

$$= \mathcal{U}(\mathbf{y}_t|\mathbf{y}_{<t}, \boldsymbol{x}). \tag{30}$$

Therefore, the uncertainty of the sequence defined in Eqn. 13:

$$\mathbb{E}_{\boldsymbol{y} \sim p(\mathbf{y}|\boldsymbol{x})}[\widetilde{\mathcal{U}}(\boldsymbol{y}|\boldsymbol{x})] = \mathbb{E}_{p(\mathbf{y}|\mathbf{x})}\left[\sum_{t=1}^{T} \mathcal{U}(\mathbf{y}_t|\boldsymbol{y}_{<t}, \boldsymbol{x})\right] \tag{31}$$

$$= \sum_{t=1}^{T} \mathbb{E}_{p(\mathbf{y}|\mathbf{x})}[\mathcal{U}(\mathbf{y}_t|\boldsymbol{y}_{<t}, \boldsymbol{x})] \tag{32}$$

$$= \sum_{t=1}^{T} \mathbb{E}_{\boldsymbol{y}_{<t} \sim p(\cdot|\boldsymbol{x})}[\mathcal{U}(\mathbf{y}_t|\boldsymbol{y}_{<t}, \boldsymbol{x})] \tag{33}$$

$$= \sum_{t=1}^{T} \mathcal{U}(\mathbf{y}_t|\mathbf{y}_{<t}, \boldsymbol{x}) \tag{34}$$

$$= \mathcal{U}(\mathbf{y}|\boldsymbol{x}), \tag{35}$$

where the final step follows from the chain rule of entropy (Proposition C.1). $\square$

**Proposition 3.2** (**Token-Level and Response-Level Uncertainty**). *Given an input query $\boldsymbol{x}$, let $\boldsymbol{y} \sim p(\mathbf{y}|\boldsymbol{x})$ be a generated sample of length $T$. Let $\mathcal{U}(\mathbf{y}_t|\boldsymbol{y}_{<t}, \boldsymbol{x})$ denote the token-level uncertainty as defined in Eqn. 10-12, with $\widetilde{\mathcal{U}}(\mathbf{y}|\boldsymbol{x})$ as the corresponding response-level uncertainty (Definition 3.1). Our token-level uncertainty is equivalent to the response-level uncertainty when $T = 1$:*

$$\widetilde{\mathcal{U}}(\boldsymbol{y}|\boldsymbol{x}) = \mathcal{U}(\mathbf{y}_1|\boldsymbol{x}). \tag{36}$$

*Proof.* When the sequence length $T = 1$, based on the definition of uncertainty of sequence in Eqn. 13, we have

$$\widetilde{\mathcal{U}}(\boldsymbol{y}|\mathrm{x}) = \sum_{t=1}^{T} \mathcal{U}(\mathbf{y}_t|\boldsymbol{y}_{<t}, \boldsymbol{x}) = \mathcal{U}(\mathbf{y}_1|\boldsymbol{x}).$$

This proposition implies that the sequence uncertainty collapses to token-level uncertainty when the output sequence length is 1, reflecting the structural consistency of the estimator. $\square$

**Proposition C.2** (**Approximate Distribution of the Weight $W$ Perturbed by Low-Rank Noise, Eqn. 18**). *Given the weight matrix $\boldsymbol{W}_0 \in \mathbb{R}^{m \times n}$, the low-rank noise matrix $\boldsymbol{\epsilon} \in \mathbb{R}^{n \times r'}$ whose rank $r' \ll r$ is significantly smaller than the rank $r$ of $\boldsymbol{W}_0$, and whose entries are sampled i.i.d. from a Gaussian distribution of standard deviation of $\sigma_q$: $\epsilon_{ij} \sim \mathcal{N}(0, \sigma_q^2), \forall i \in [n], j \in [r']$, we have the perturbed weighted matrix $\boldsymbol{W}$ as defined in Eqn. 17. The variational distribution $q(\mathrm{vec}(\boldsymbol{W})|\sigma_q)$ defined on the weight matrix $\boldsymbol{W}$ is*

$$q(\mathrm{vec}(\boldsymbol{W})|\sigma_q) = \mathcal{N}(\mathrm{vec}(\boldsymbol{W})|\boldsymbol{\mu}_q, \boldsymbol{\Sigma}_q),$$
$$where \quad \boldsymbol{\mu}_q = \mathrm{vec}(\boldsymbol{W}_0),$$
$$\boldsymbol{\Sigma}_q = \sigma_q^2 \boldsymbol{I}_n \otimes \begin{bmatrix} \boldsymbol{I}_{r'} & \\ & \boldsymbol{0}_{m-r'} \end{bmatrix}. \tag{37}$$

*Proof.* We begin with compact SVD decomposition of the weight matrix $\boldsymbol{W}_0$ as described in Eqn. 16:

$$\boldsymbol{W}_0 = \boldsymbol{U}\mathrm{diag}(\boldsymbol{d})\boldsymbol{V}^\top, \tag{38}$$

where $\boldsymbol{d} \succ \boldsymbol{0} \in \mathbb{R}^{r \times 1}$ is the vector of singular values, and $\boldsymbol{U} \in \mathbb{R}^{m \times r}$, $\boldsymbol{V} \in \mathbb{R}^{n \times r}$ are orthogonal matrices. We denote the first $r'$ columns of $\boldsymbol{U}$ as $\boldsymbol{U}' \in \mathbb{R}^{m \times r'}$ to analyze the updated matrix $\boldsymbol{U}'\boldsymbol{\epsilon}^\top$ in Eqn. 17.

Since each entry in $\boldsymbol{\epsilon}$ has zero mean, it is evident that the updated matrix also has zero mean. Consequently, we have $\boldsymbol{\mu}_q = \mathrm{vec}(\boldsymbol{W}_0) + \boldsymbol{0} = \mathrm{vec}(\boldsymbol{W}_0)$.

Next, we focus on the proof of the variance $\boldsymbol{\Sigma}_q$. Gien $\boldsymbol{U}' = (\boldsymbol{u}_1, \boldsymbol{u}_2, \cdots, \boldsymbol{u}_{r'}) \in \mathbb{R}^{m \times r'}$, and $\boldsymbol{\epsilon} = (\boldsymbol{\epsilon}_1, \boldsymbol{\epsilon}_2, \cdots, \boldsymbol{\epsilon}_{r'}) \in \mathbb{R}^{n \times r'}$ as defined above, we have the following properties:

$$\boldsymbol{U}'\boldsymbol{U}'^\top = \sum_{i=1}^{r'} \boldsymbol{u}_i \boldsymbol{u}_i^\top = \begin{bmatrix} \boldsymbol{I}_{r'} & \\ & \boldsymbol{0}_{m-r'} \end{bmatrix}, \tag{39}$$

$$\mathrm{vec}(\boldsymbol{U}'\boldsymbol{\epsilon}^\top) = \mathrm{vec}(\sum_{i=1}^{r'} \boldsymbol{u}_i \boldsymbol{\epsilon}_i^\top) = \sum_{i=1}^{r'} (\boldsymbol{\epsilon}_i \otimes \boldsymbol{u}_i). \tag{40}$$

We can now derive the covariance matrix as:

$$\boldsymbol{\Sigma}_q = \mathrm{Var}[\mathrm{vec}(\boldsymbol{W})] = \mathrm{Var}[\mathrm{vec}(\boldsymbol{W}_0 + \boldsymbol{U}'\boldsymbol{\epsilon}^\top)] = \mathrm{Var}[\mathrm{vec}(\boldsymbol{U}'\boldsymbol{\epsilon}^\top)] \tag{41}$$

$$= \mathrm{Var}[\sum_{i=1}^{r'} \boldsymbol{\epsilon}_i \otimes \boldsymbol{u}_i] = \sum_{i=1}^{r'} \mathrm{Var}[\boldsymbol{\epsilon}_i \otimes \boldsymbol{u}_i] \tag{42}$$

$$= \sum_{i=1}^{r'} \left\{ \mathbb{E}_{\boldsymbol{\epsilon}_i}[(\boldsymbol{\epsilon}_i \otimes \boldsymbol{u}_i)(\boldsymbol{\epsilon}_i \otimes \boldsymbol{u}_i)^\top] - \mathbb{E}_{\boldsymbol{\epsilon}_i}[(\boldsymbol{\epsilon}_i \otimes \boldsymbol{u}_i)]\mathbb{E}_{\boldsymbol{\epsilon}_i}[(\boldsymbol{\epsilon}_i \otimes \boldsymbol{u}_i)^\top] \right\} \tag{43}$$

$$= \sum_{i=1}^{r'} \left\{ \mathbb{E}_{\boldsymbol{\epsilon}_i}[\boldsymbol{\epsilon}_i \boldsymbol{\epsilon}_i^\top] \otimes (\boldsymbol{u}_i \boldsymbol{u}_i^\top) - (\mathbb{E}_{\boldsymbol{\epsilon}_i}[\boldsymbol{\epsilon}_i]\mathbb{E}_{\boldsymbol{\epsilon}_i}[\boldsymbol{\epsilon}_i]^\top) \otimes (\boldsymbol{u}_i \boldsymbol{u}_i^\top) \right\} \tag{44}$$

$$= \sum_{i=1}^{r'} \sigma_q^2 \boldsymbol{I}_n \otimes (\boldsymbol{u}_i \boldsymbol{u}_i^\top) = \sigma_q^2 \boldsymbol{I}_n \otimes \sum_{i=1}^{r'} \boldsymbol{u}_i \boldsymbol{u}_i^\top = \sigma_q^2 \boldsymbol{I}_n \otimes \begin{bmatrix} \boldsymbol{I}_{r'} & \\ & \boldsymbol{0}_{m-r'} \end{bmatrix}. \tag{45}$$

$\square$

# D IMPLEMENTATION DETAILS

## D.1 IMPLEMENTATION OF TOKUR'S TOKEN-LEVEL UNCERTAINTIES

Unless otherwise specified, we set the rank of low-rank noise to $r' = 8$, the perturbation strength $\sigma_q = 0.1$, and the number of samples per uncertainty estimation to $M = 2$. For the test-time scaling experiments in Sec. 4.2, we apply length normalization to TOKUR to mitigate the bias introduced by varying sequence lengths. In contrast, the effect of length normalization may differ in hallucination detection tasks. To investigate this, we conduct additional ablation studies in Appendix E.5.3, examining the impact of length normalization in that setting. To ensure practical applicability in real-world scenarios, we implement our method as a seamless integration with vLLM (Kwon et al., 2023).

## D.2 DATASETS

Table 5 shows the statistics of datasets in our experiments. These datasets collectively span a wide range of difficulty levels, from moderate to highly challenging, covering both elementary-level numerical reasoning and advanced symbolic mathematical tasks. In addition, the problem domains are diverse, including: algebra, geometry, and number theory. Such a design ensures that our experiments are comprehensive and representative, facilitating a thorough assessment of the model's capability across varied reasoning scenarios.

Table 5: Statistics of the datasets used in our experiments.

| Dataset | Samples Used | Split | Task Type | Language | Level |
|---------|--------------|-------|-----------|----------|-------|
| GSM8K | 1,300 | Training split | Mathematical Reasoning | English | Moderate |
| MATH500 | 500 | Full set | Mathematical Reasoning | English | Difficult |
| DeepScaleR | 5,000 | First 5,000 samples | Mathematical Reasoning | English | Highly Challenging |

### D.3 PROMPT TEMPLATES

In this work, we use the following prompts published by Meta [2].

---
**Prompt Example**

Solve the following math problem efficiently and clearly:

-For simple problems (2 steps or fewer):
Provide a concise solution with minimal explanation.

-For complex problems (3 steps or more):
Use this step-by-step format:

    ## Step 1: [Concise description]
    [Brief explanation and calculations]

    ## Step 2: [Concise description]
    [Brief explanation and calculations]
...

Regardless of the approach, always conclude with:
Therefore, the final answer is: $\boxed{answer}$. I hope it is correct.
Where [answer] is just the final number or expression that solves the problem.
---

### D.4 BASELINES

We compare our uncertainty estimation approach against several baseline methods:

- **Log-Likelihood (LL)** (Murray & Chiang, 2018): Mean of token-wise log-probabilities of the output sequence, representing the model's overall confidence in its generation.
- **Predictive Entropy (PE)** (Malinin & Gales, 2021): Mean entropy of the predicted distribution of each token.
- **P(True)** (Kadavath et al., 2022): Directly queries the model about the correctness of its own output and uses the predicted probability of the token "True", normalized by the sum of probabilities of token "True" and "False", as a confidence score.
- **Self-Certainty** (Kang et al., 2025): Quantifies confidence using the KL divergence between the predicted token distribution and a uniform distribution over the vocabulary at each decoding step.
- **DeepConf** (Fu et al., 2025): Computes confidence scores by aggregating the log-probabilities of the top-$K$ candidate tokens at each decoding step.
- **The Degree Matrix** (Lin et al., 2023): Utilizes the degree matrix of the graph Laplacian of the similarities matrix of responses.
- **LLM-Check** (Sriramanan et al., 2024): We faithfully reproduced the official implementation for comparison.
- **INSIDE** (Chen et al., 2024a): INSIDE is a method to estimate query-level uncertainty. We tailored INSIDE to our setting by asking the LLM to verify the same response multiple times and then calculating the semantic entropy across these verification attempts.

---
[2]https://huggingface.co/datasets/meta-llama/Llama-3.2-1B-Instruct-evals

- **Semantic Entropy (SE)** (Kuhn et al., 2023): We adapted SE to our setting by prompting the LLM to verify the same response multiple times and computing the semantic entropy of these verification attempts. While this provides a signal of response quality, we note that SE requires an external NLI or embedding model, giving it an inherent advantage compared to our method.
- **SAR** (Duan et al., 2024): We adapted SAR to our setting by computing sentence-level SAR scores over multiple verification attempts, following a similar procedure to SE. While this method provides a meaningful proxy for uncertainty, **it requires an external semantic similarity model**, which raises fairness concerns compared to our approach, which operates solely with the base LLM.

### D.5 EVALUATION PARSING AND METRICS

**Parsing.** To automate the evaluation of outputs generated by large language models, we design specific prompts (see Appendix D.3) that constrain the model to follow a fixed structure and require it to place the final answer within a \box{}. Considering that in mathematical reasoning tasks, the same answer can be expressed in various forms, we standardize all answers into a canonical form before comparison (Beeching et al.). During the evaluation, we assess the correctness from two perspectives: numerical equality and symbolic equality, to label each generation as "True" or "False".

**Metrics.** To comprehensively assess model performance in binary classification tasks, we adopt the following metrics: Area Under the Receiver Operating Characteristic Curve (AUROC), Area Under the Precision-Recall Curve (AUPRC), and Top 50% Accuracy (Farquhar et al., 2024; Ye et al., 2025; Hanley & McNeil, 1982; Boyd et al., 2013).

- **AUROC** measures the trade-off between true positive rate (TPR) and false positive rate (FPR) at various threshold settings. Formally, for a set of predictions with associated confidence scores, AUROC is computed as:

$$\text{AUROC} = \int_0^1 \text{TPR}(\text{FPR}^{-1}(x))\, dx, \tag{46}$$

where TPR and FPR are defined as:

$$\text{TPR} = \tfrac{\text{TP}}{\text{TP+FN}}, \quad \text{FPR} = \tfrac{\text{FP}}{\text{FP+TN}}.$$

- **AUPRC** evaluates the trade-off between precision and recall, which is particularly useful in imbalanced datasets. It is calculated as:

$$\text{AUPRC} = \int_0^1 \text{Precision}(\text{Recall}^{-1}(x))\, dx, \tag{47}$$

where precision and recall are defined as:

$$\text{Precision} = \tfrac{\text{TP}}{\text{TP+FP}}, \quad \text{Recall} = \tfrac{\text{TP}}{\text{TP+FN}}.$$

- **Top 50% Accuracy** evaluates the correctness of the top half predictions ranked by confidence. Let $N$ be the total number of predictions and $S$ be the set of indices corresponding to the top $\lceil N/2 \rceil$ predictions with highest confidence. The metric is defined as:

$$\text{Top 50\% Accuracy} = \tfrac{1}{|S|} \sum_{i \in S} \delta(\widehat{y}_i = y_i), \tag{48}$$

where $\widehat{y}_i$ is the predicted label and $y_i$ is the ground-truth label.

## E ADDITIONAL EXPERIMENTAL RESULTS

### E.1 PRELIMINARY STUDY: DISTRIBUTION OF UNCERTAINTIES

We conduct a preliminary study to examine the relationship between responses' token-level uncertainties and their correctness. We generate responses on the GSM8K dataset using a greedy decoding strategy with `Llama-3.2-1B-Instruct`, and label each response as correct or incorrect based

Table 6: **Performance of Uncertainty Estimation Methods for Incorrect Reasoning Path Detection of Qwen famliy models.** AUROC, AUPRC, and ACC* are all reported as percentage (%). We include the accuracy of CoT (i.e., greedy decoding with Chain-of-Thought prompting) in the first row for reference. **Boldface** and underlining denote the best and the second-best performance, respectively.

| Method | MATH500 | | | GSM8K | | | DeepScaleR | | |
|---|---|---|---|---|---|---|---|---|---|
| | AUROC | AUPRC | ACC* | AUROC | AUPRC | ACC* | AUROC | AUPRC | ACC* |
| Qwen-2.5-3B-Instruct | | | | | | | | | |
| CoT (Lower-Bound) | - | - | $64.73_{\pm0.00}$ | - | - | $87.72_{\pm0.00}$ | - | - | $35.87_{\pm0.00}$ |
| PE | $68.94_{\pm0.27}$ | $77.30_{\pm0.62}$ | $76.80_{\pm0.40}$ | $76.44_{\pm0.10}$ | $95.06_{\pm0.01}$ | $94.46_{\pm0.00}$ | $66.57_{\pm0.07}$ | $46.44_{\pm0.02}$ | $48.09_{\pm0.02}$ |
| LL | $68.42_{\pm0.29}$ | $76.82_{\pm0.79}$ | $76.13_{\pm0.23}$ | $75.54_{\pm0.12}$ | $94.93_{\pm0.01}$ | $94.92_{\pm0.00}$ | $65.66_{\pm0.06}$ | $45.43_{\pm0.02}$ | $47.37_{\pm0.02}$ |
| Self-Certainty | $74.95_{\pm0.40}$ | $84.76_{\pm0.40}$ | $81.33_{\pm0.61}$ | $77.22_{\pm0.00}$ | $\underline{95.66}_{\pm0.02}$ | $94.77_{\pm0.00}$ | $74.22_{\pm0.08}$ | $62.55_{\pm0.02}$ | $52.00_{\pm0.00}$ |
| DeepConf | $74.39_{\pm0.33}$ | $83.81_{\pm0.39}$ | $80.53_{\pm0.61}$ | $\underline{78.06}_{\pm0.01}$ | $95.63_{\pm0.03}$ | $95.69_{\pm0.00}$ | $73.85_{\pm0.07}$ | $60.75_{\pm0.02}$ | $51.93_{\pm0.02}$ |
| TokUR (TU, Ours) | $\underline{81.48}_{\pm0.09}$ | $\underline{87.42}_{\pm0.09}$ | $\underline{86.67}_{\pm0.23}$ | $77.66_{\pm0.40}$ | $95.46_{\pm0.12}$ | $\underline{96.05}_{\pm0.18}$ | $\underline{76.26}_{\pm0.02}$ | $\underline{65.47}_{\pm0.07}$ | $\underline{53.52}_{\pm0.14}$ |
| TokUR (AU, Ours) | $\mathbf{82.55}_{\pm0.05}$ | $\mathbf{88.30}_{\pm0.15}$ | $\mathbf{87.33}_{\pm0.23}$ | $\mathbf{78.81}_{\pm0.31}$ | $\mathbf{95.76}_{\pm0.16}$ | $\mathbf{96.15}_{\pm0.31}$ | $\mathbf{77.52}_{\pm0.03}$ | $\mathbf{67.30}_{\pm0.08}$ | $\mathbf{54.61}_{\pm0.27}$ |
| TokUR (EU, Ours) | $78.96_{\pm0.13}$ | $85.37_{\pm0.24}$ | $84.27_{\pm0.46}$ | $70.75_{\pm0.81}$ | $93.60_{\pm0.21}$ | $94.26_{\pm0.44}$ | $73.33_{\pm0.02}$ | $61.08_{\pm0.13}$ | $51.35_{\pm0.10}$ |
| Qwen-2.5-7B-Instruct | | | | | | | | | |
| CoT (Lower-Bound) | - | - | $75.07_{\pm0.00}$ | - | - | $92.15_{\pm0.00}$ | - | - | $44.93_{\pm0.00}$ |
| PE | $62.51_{\pm0.37}$ | $80.29_{\pm0.45}$ | $82.00_{\pm0.40}$ | $77.40_{\pm0.19}$ | $97.35_{\pm0.01}$ | $97.23_{\pm0.00}$ | $60.55_{\pm0.09}$ | $50.61_{\pm0.10}$ | $52.64_{\pm0.14}$ |
| LL | $62.19_{\pm0.39}$ | $80.19_{\pm0.46}$ | $82.13_{\pm0.61}$ | $75.37_{\pm0.22}$ | $97.02_{\pm0.00}$ | $97.08_{\pm0.00}$ | $60.23_{\pm0.10}$ | $50.27_{\pm0.11}$ | $52.43_{\pm0.14}$ |
| Self-Certainty | $69.54_{\pm0.06}$ | $86.24_{\pm0.10}$ | $85.60_{\pm0.40}$ | $\underline{80.92}_{\pm0.15}$ | $97.81_{\pm0.01}$ | $97.69_{\pm0.00}$ | $66.31_{\pm0.07}$ | $58.18_{\pm0.10}$ | $56.60_{\pm0.08}$ |
| DeepConf | $66.86_{\pm0.06}$ | $83.85_{\pm0.03}$ | $84.53_{\pm0.23}$ | $\mathbf{81.94}_{\pm0.15}$ | $\mathbf{97.88}_{\pm0.00}$ | $\mathbf{98.31}_{\pm0.00}$ | $64.91_{\pm0.05}$ | $56.71_{\pm0.08}$ | $55.83_{\pm0.08}$ |
| TokUR (TU, Ours) | $\underline{84.61}_{\pm0.62}$ | $\underline{93.43}_{\pm0.25}$ | $\mathbf{94.00}_{\pm0.40}$ | $80.15_{\pm0.12}$ | $97.36_{\pm0.11}$ | $97.69_{\pm0.00}$ | $\underline{76.05}_{\pm0.12}$ | $\underline{71.43}_{\pm0.31}$ | $\underline{63.89}_{\pm0.33}$ |
| TokUR (AU, Ours) | $\mathbf{84.74}_{\pm0.52}$ | $\mathbf{93.65}_{\pm0.17}$ | $\mathbf{94.00}_{\pm0.40}$ | $80.70_{\pm0.10}$ | $97.44_{\pm0.14}$ | $\underline{97.79}_{\pm0.09}$ | $\mathbf{76.39}_{\pm0.10}$ | $\mathbf{71.73}_{\pm0.31}$ | $\mathbf{64.24}_{\pm0.30}$ |
| TokUR (EU, Ours) | $81.87_{\pm1.02}$ | $91.68_{\pm0.35}$ | $\underline{92.00}_{\pm1.06}$ | $74.95_{\pm0.40}$ | $96.61_{\pm0.08}$ | $97.23_{\pm0.15}$ | $73.55_{\pm0.21}$ | $68.91_{\pm0.48}$ | $61.47_{\pm0.13}$ |

on an exact match with the ground-truth answer. Considering the class imbalance in the model responses, we construct a balanced subset for visualization. Specifically, we retain all incorrect responses and randomly sample an equal number of correct responses. **We compute the TokUR (EU) and TokUR (AU) with our proposed token-level uncertainties in Eqn. 13, and plot the results in the Normalized EU-AU space (Fig. 3).** We observe that both TokUR (EU) and TokUR (AU) show a better-than-chance separation between correct and incorrect outputs. Although some overlap exists, their distribution peaks differ significantly, indicating that our uncertainty estimates meaningfully correlate with generation quality.

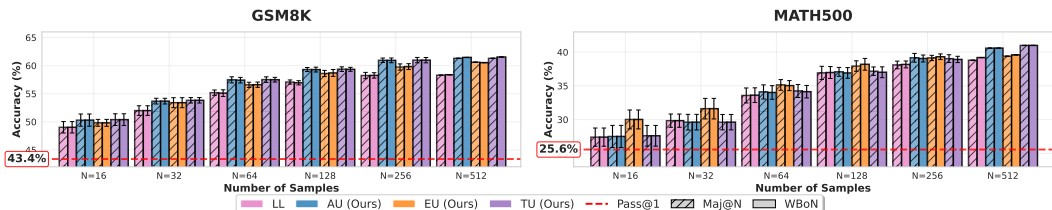

Figure 2: **Performance on GSM8K (Left) and MATH500 (Right) when scaling up sample size $N$ at test time of Llama-3.2-1B-Instruct.** Our TokUR (AU, EU, and TU) consistently outperforms the LL baseline, particularly when $N$ is small. Please refer to Table 7 for detailed numerical results.

## E.2 UNCERTAINTY ESTIMATION FOR QWEN MODEL

As described in Sec. 4.1.2, we further evaluate the generalization of TokUR by conducting experiments with the Qwen models. **Table 6 reports results for Qwen-2.5-3B-Instruct and Qwen-2.5-7B-Instruct,** using the same experimental setup as in Table 1. Overall, these results provide strong additional evidence that TokUR's token-level uncertainty estimates maintain a robust correlation with model accuracy across diverse LLM families and parameter scales.

## E.3 TEST-TIME SCALING VIA UNCERTAINTY ESTIMATION

**We provide an additional visualization of the test-time scaling results in Fig. 2 .** While the complete numerical results are reported in Table 4, this new figure offers an intuitive view of how accuracy improves with increasing numbers of test-time samples ($N \in 16, 32, 64, 128, 256, 512$).

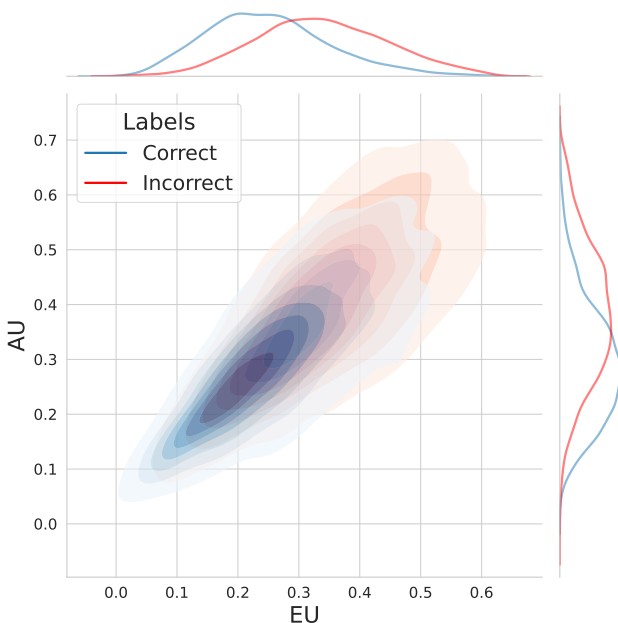

Figure 3: Distribution of responses from GSM8K (Cobbe et al., 2021) plotted in the Length Normalized EU-AU uncertainty space, as quantified by our token-level uncertainty metrics (Eqn. 13).

All experiments use `Llama-3.2-1B-Instruct` as the base model. For reference, the Pass@1 baseline accuracy (GSM8K: 44.43%; MATH500: 25.60%) is also shown as red dashed lines, highlighting the gains achieved through test-time scaling. Moreover, **we provide an extended version of the results in Table 7**, which builds on Table 4 to include additional test-time sample configurations.

### E.4 TOKUR FOR TEST-TIME SCALING (ONLINE)

One popular approach to improving model performance uses a Process Reward Model (PRM) to score each intermediate step during multi-step generation (Guan et al., 2025; Puri et al., 2025; Beeching et al.; Uesato et al., 2022; Lightman et al., 2023), thereby guiding the model's reasoning path. In this section, we explore an alternative: guiding the generation process using uncertainty as an intrinsic reward, without relying on an explicit reward model.

**Experimental Setting.** Particle Filtering (PF) (Puri et al., 2025) is an inference-time scaling method for LLM reasoning (details in Appendix B). Building upon this algorithm, we use uncertainty as the score for each particle at each step to guide the model's generation process. We set the number of particles to $N = 16$ and the decoding temperature to $\tau = 0.8$. We repeat the experiments with three different random seeds to obtain the mean and standard deviation across runs.

**Results.** Table 8 shows the results. Compared to LL, our `TokUR`, especially `TokUR` (EU), yields a slight performance gain. Given that guiding generation through stepwise scoring is inherently challenging, we consider the lack of a significant performance gain from uncertainty estimation to be acceptable. Nevertheless, we believe this experiment offers valuable insights that may inform the future design of process reward models.

Table 8: **`TokUR` as Implicit Reward for Test-Time Scaling (Online), on MATH500.**

| Intrinsic Reward | BoN | WBoN |
|---|---|---|
| LL | $26.27_{\pm 0.25}$ | $26.27_{\pm 0.41}$ |
| `TokUR` (TU, Ours) | $27.93_{\pm 0.25}$ | $28.13_{\pm 0.38}$ |
| `TokUR` (AU, Ours) | $25.20_{\pm 0.99}$ | $25.13_{\pm 0.74}$ |
| `TokUR` (EU, Ours) | $\mathbf{28.93_{\pm 0.08}}$ | $\mathbf{29.20_{\pm 0.98}}$ |

Table 7: **Test-Time Scaling for GSM8K and MATH500.** Performance comparison of different methods with varying numbers of test-time samples (N = 16 to 512) using Llama-3.2-1B-Instruct as the base model. Methods evaluated include log-likelihood (LL) and three variants of TokUR (TU, AU and EU) with both Maj@N and WBoN strategies. **Boldface** and underlining denote the best and the second-best performance, respectively.

| Dataset | Score | Method | Number of Samples N | | | | | |
|---|---|---|---|---|---|---|---|---|
| | | | N=16 | N=32 | N=64 | N=128 | N=256 | N=512 |
| | | | Llama-3.2-1B-Instruct | | | | | |
| **GSM8K** (Pass@1: 44.43) | LL | Maj@N | $47.10_{\pm 0.85}$ | $50.45_{\pm 0.64}$ | $54.11_{\pm 0.52}$ | $56.77_{\pm 0.40}$ | $58.89_{\pm 0.36}$ | $59.72_{\pm 0.00}$ |
| | | WBoN | $47.10_{\pm 0.85}$ | $50.45_{\pm 0.64}$ | $54.15_{\pm 0.55}$ | $56.72_{\pm 0.42}$ | $58.92_{\pm 0.37}$ | $59.81_{\pm 0.00}$ |
| | Self-Certainty | Maj@N | $45.02_{\pm 0.92}$ | $48.97_{\pm 0.81}$ | $52.61_{\pm 0.72}$ | $55.22_{\pm 0.67}$ | $57.18_{\pm 0.53}$ | $58.03_{\pm 0.00}$ |
| | | WBoN | $45.02_{\pm 0.92}$ | $48.97_{\pm 0.81}$ | $52.65_{\pm 0.70}$ | $55.30_{\pm 0.64}$ | $57.22_{\pm 0.54}$ | $58.10_{\pm 0.00}$ |
| | DeepConf | Maj@N | $46.72_{\pm 0.89}$ | $50.12_{\pm 0.71}$ | $53.50_{\pm 0.66}$ | $56.10_{\pm 0.52}$ | $58.05_{\pm 0.44}$ | $58.97_{\pm 0.00}$ |
| | | WBoN | $46.72_{\pm 0.89}$ | $50.12_{\pm 0.71}$ | $53.47_{\pm 0.65}$ | $56.08_{\pm 0.49}$ | $58.10_{\pm 0.45}$ | $59.05_{\pm 0.00}$ |
| | TokUR (TU, Ours) | Maj@N | $\underline{50.29}_{\pm 1.03}$ | $\underline{53.72}_{\pm 0.77}$ | $57.18_{\pm 0.45}$ | $59.10_{\pm 0.60}$ | $\underline{60.68}_{\pm 0.49}$ | $61.23_{\pm 0.00}$ |
| | | WBoN | $\underline{50.29}_{\pm 1.03}$ | $\underline{53.72}_{\pm 0.77}$ | $\mathbf{57.22}_{\pm 0.45}$ | $\mathbf{59.21}_{\pm 0.66}$ | $\underline{60.71}_{\pm 0.49}$ | $\underline{61.31}_{\pm 0.00}$ |
| | TokUR (AU, Ours) | Maj@N | $50.20_{\pm 0.98}$ | $\mathbf{53.77}_{\pm 0.90}$ | $\underline{57.21}_{\pm 0.46}$ | $58.99_{\pm 0.61}$ | $60.70_{\pm 0.41}$ | $\mathbf{61.38}_{\pm 0.00}$ |
| | | WBoN | $50.20_{\pm 0.98}$ | $\mathbf{53.77}_{\pm 0.90}$ | $57.19_{\pm 0.44}$ | $\underline{59.13}_{\pm 0.67}$ | $\mathbf{60.78}_{\pm 0.42}$ | $\underline{61.31}_{\pm 0.00}$ |
| | TokUR (EU, Ours) | Maj@N | $\mathbf{50.38}_{\pm 0.92}$ | $52.98_{\pm 0.67}$ | $56.92_{\pm 0.60}$ | $58.77_{\pm 0.38}$ | $59.88_{\pm 0.52}$ | $60.69_{\pm 0.00}$ |
| | | WBoN | $\mathbf{50.38}_{\pm 0.92}$ | $52.98_{\pm 0.67}$ | $56.89_{\pm 0.54}$ | $58.70_{\pm 0.40}$ | $59.91_{\pm 0.58}$ | $60.85_{\pm 0.00}$ |
| **MATH500** (Pass@1: 25.60) | LL | Maj@N | $26.42_{\pm 0.84}$ | $29.28_{\pm 0.89}$ | $33.28_{\pm 0.97}$ | $37.04_{\pm 0.74}$ | $38.56_{\pm 0.75}$ | $39.00_{\pm 0.00}$ |
| | | WBoN | $26.42_{\pm 0.84}$ | $\underline{29.28}_{\pm 0.89}$ | $33.30_{\pm 1.10}$ | $37.02_{\pm 0.84}$ | $38.58_{\pm 0.73}$ | $39.00_{\pm 0.00}$ |
| | Self-Certainty | Maj@N | $20.14_{\pm 1.14}$ | $23.24_{\pm 1.44}$ | $29.12_{\pm 1.11}$ | $33.80_{\pm 0.89}$ | $36.68_{\pm 0.83}$ | $38.60_{\pm 0.00}$ |
| | | WBoN | $20.14_{\pm 1.14}$ | $23.24_{\pm 1.44}$ | $29.16_{\pm 0.99}$ | $33.82_{\pm 0.82}$ | $36.80_{\pm 0.80}$ | $38.60_{\pm 0.00}$ |
| | DeepConf | Maj@N | $25.68_{\pm 1.38}$ | $28.36_{\pm 0.91}$ | $33.30_{\pm 1.10}$ | $37.34_{\pm 1.31}$ | $38.52_{\pm 0.43}$ | $\mathbf{40.40}_{\pm 0.00}$ |
| | | WBoN | $25.68_{\pm 1.38}$ | $28.08_{\pm 1.08}$ | $32.44_{\pm 1.20}$ | $36.00_{\pm 1.22}$ | $37.08_{\pm 0.78}$ | $38.60_{\pm 0.00}$ |
| | TokUR (TU, Ours) | Maj@N | $\underline{27.06}_{\pm 0.94}$ | $29.18_{\pm 1.06}$ | $\underline{33.76}_{\pm 0.84}$ | $37.62_{\pm 0.70}$ | $39.18_{\pm 0.70}$ | $39.00_{\pm 0.00}$ |
| | | WBoN | $\underline{27.06}_{\pm 0.94}$ | $29.18_{\pm 1.06}$ | $33.60_{\pm 0.82}$ | $37.60_{\pm 0.79}$ | $39.20_{\pm 0.65}$ | $39.40_{\pm 0.00}$ |
| | TokUR (AU, Ours) | Maj@N | $27.06_{\pm 0.91}$ | $29.08_{\pm 1.14}$ | $33.64_{\pm 0.76}$ | $37.52_{\pm 0.82}$ | $39.12_{\pm 0.72}$ | $39.20_{\pm 0.00}$ |
| | | WBoN | $\underline{27.06}_{\pm 0.91}$ | $29.08_{\pm 1.14}$ | $33.48_{\pm 0.73}$ | $37.64_{\pm 0.73}$ | $39.10_{\pm 0.69}$ | $39.60_{\pm 0.00}$ |
| | TokUR (EU, Ours) | Maj@N | $\mathbf{28.28}_{\pm 1.32}$ | $\mathbf{31.36}_{\pm 1.05}$ | $\mathbf{35.44}_{\pm 0.79}$ | $\mathbf{38.00}_{\pm 0.77}$ | $\mathbf{39.44}_{\pm 0.88}$ | $39.60_{\pm 0.00}$ |
| | | WBoN | $\mathbf{28.28}_{\pm 1.32}$ | $\mathbf{31.36}_{\pm 1.05}$ | $\mathbf{35.44}_{\pm 0.78}$ | $\underline{37.86}_{\pm 0.84}$ | $\underline{39.38}_{\pm 0.87}$ | $40.00_{\pm 0.00}$ |
| | | | Llama-3.1-8B-Instruct | | | | | |
| **GSM8K** (Pass@1: 85.69) | LL | Maj@N | $86.74_{\pm 0.62}$ | $89.16_{\pm 0.53}$ | $90.48_{\pm 0.48}$ | $90.99_{\pm 0.35}$ | $91.01_{\pm 0.28}$ | $91.00_{\pm 0.00}$ |
| | | WBoN | $86.74_{\pm 0.62}$ | $89.16_{\pm 0.53}$ | $90.48_{\pm 0.49}$ | $90.99_{\pm 0.36}$ | $91.00_{\pm 0.29}$ | $91.00_{\pm 0.00}$ |
| | Self-Certainty | Maj@N | $80.02_{\pm 0.70}$ | $84.13_{\pm 0.66}$ | $87.25_{\pm 0.49}$ | $89.22_{\pm 0.40}$ | $90.05_{\pm 0.40}$ | $90.77_{\pm 0.00}$ |
| | | WBoN | $80.02_{\pm 0.70}$ | $84.13_{\pm 0.66}$ | $87.25_{\pm 0.50}$ | $89.21_{\pm 0.39}$ | $90.05_{\pm 0.41}$ | $90.77_{\pm 0.00}$ |
| | DeepConf | Maj@N | $86.24_{\pm 0.66}$ | $88.74_{\pm 0.64}$ | $90.34_{\pm 0.46}$ | $90.88_{\pm 0.46}$ | $90.92_{\pm 0.28}$ | $91.01_{\pm 0.00}$ |
| | | WBoN | $86.24_{\pm 0.66}$ | $88.74_{\pm 0.64}$ | $90.32_{\pm 0.46}$ | $90.90_{\pm 0.45}$ | $90.94_{\pm 0.28}$ | $91.02_{\pm 0.00}$ |
| | TokUR (TU, Ours) | Maj@N | $\underline{87.68}_{\pm 0.57}$ | $\underline{89.72}_{\pm 0.55}$ | $\underline{90.67}_{\pm 0.45}$ | $\underline{91.06}_{\pm 0.38}$ | $90.96_{\pm 0.36}$ | $91.02_{\pm 0.00}$ |
| | | WBoN | $\underline{87.68}_{\pm 0.57}$ | $\underline{89.72}_{\pm 0.55}$ | $90.65_{\pm 0.46}$ | $\underline{91.06}_{\pm 0.37}$ | $90.98_{\pm 0.37}$ | $\underline{91.02}_{\pm 0.00}$ |
| | TokUR (AU, Ours) | Maj@N | $87.42_{\pm 0.66}$ | $89.59_{\pm 0.55}$ | $90.60_{\pm 0.44}$ | $91.01_{\pm 0.31}$ | $\underline{90.99}_{\pm 0.32}$ | $90.93_{\pm 0.00}$ |
| | | WBoN | $87.42_{\pm 0.66}$ | $89.59_{\pm 0.55}$ | $90.57_{\pm 0.43}$ | $91.04_{\pm 0.35}$ | $90.98_{\pm 0.30}$ | $\underline{91.02}_{\pm 0.00}$ |
| | TokUR (EU, Ours) | Maj@N | $\mathbf{88.06}_{\pm 0.57}$ | $\mathbf{89.88}_{\pm 0.39}$ | $\mathbf{90.69}_{\pm 0.47}$ | $\mathbf{91.19}_{\pm 0.40}$ | $\mathbf{91.07}_{\pm 0.33}$ | $\underline{91.02}_{\pm 0.00}$ |
| | | WBoN | $\mathbf{88.06}_{\pm 0.57}$ | $\mathbf{89.88}_{\pm 0.39}$ | $\underline{90.67}_{\pm 0.48}$ | $\mathbf{91.19}_{\pm 0.39}$ | $\mathbf{91.09}_{\pm 0.36}$ | $\mathbf{91.05}_{\pm 0.00}$ |
| **MATH500** (Pass@1: 48.60) | LL | Maj@N | $50.92_{\pm 1.77}$ | $55.24_{\pm 0.51}$ | $59.36_{\pm 0.74}$ | $62.86_{\pm 0.70}$ | $64.10_{\pm 0.61}$ | $65.00_{\pm 0.00}$ |
| | | WBoN | $50.92_{\pm 1.77}$ | $55.24_{\pm 0.51}$ | $59.46_{\pm 0.78}$ | $62.80_{\pm 0.78}$ | $64.02_{\pm 0.71}$ | $65.00_{\pm 0.00}$ |
| | Self-Certainty | Maj@N | $44.00_{\pm 1.82}$ | $48.48_{\pm 1.06}$ | $55.56_{\pm 1.08}$ | $60.04_{\pm 0.56}$ | $62.66_{\pm 0.75}$ | $65.40_{\pm 0.00}$ |
| | | WBoN | $44.00_{\pm 1.82}$ | $48.48_{\pm 1.06}$ | $55.58_{\pm 1.06}$ | $59.90_{\pm 0.51}$ | $62.52_{\pm 0.53}$ | $64.80_{\pm 0.00}$ |
| | DeepConf | Maj@N | $49.88_{\pm 1.29}$ | $55.04_{\pm 1.44}$ | $59.74_{\pm 1.17}$ | $62.40_{\pm 0.63}$ | $64.30_{\pm 0.63}$ | $65.20_{\pm 0.00}$ |
| | | WBoN | $49.88_{\pm 1.29}$ | $54.42_{\pm 1.46}$ | $58.22_{\pm 1.00}$ | $60.90_{\pm 1.02}$ | $63.14_{\pm 0.55}$ | $64.80_{\pm 0.00}$ |
| | TokUR (TU, Ours) | Maj@N | $\underline{51.26}_{\pm 1.36}$ | $\underline{55.54}_{\pm 0.70}$ | $59.44_{\pm 1.31}$ | $62.28_{\pm 0.95}$ | $63.86_{\pm 0.44}$ | $65.20_{\pm 0.00}$ |
| | | WBoN | $\underline{51.26}_{\pm 1.36}$ | $\underline{55.54}_{\pm 0.70}$ | $59.44_{\pm 1.30}$ | $62.32_{\pm 1.08}$ | $63.84_{\pm 0.51}$ | $65.20_{\pm 0.00}$ |
| | TokUR (AU, Ours) | Maj@N | $51.16_{\pm 1.45}$ | $55.52_{\pm 0.66}$ | $59.42_{\pm 1.16}$ | $62.32_{\pm 1.07}$ | $64.00_{\pm 0.44}$ | $\underline{65.60}_{\pm 0.00}$ |
| | | WBoN | $51.16_{\pm 1.45}$ | $55.52_{\pm 0.66}$ | $59.44_{\pm 1.19}$ | $62.34_{\pm 1.16}$ | $63.92_{\pm 0.47}$ | $\underline{65.60}_{\pm 0.00}$ |
| | TokUR (EU, Ours) | Maj@N | $\mathbf{52.40}_{\pm 1.39}$ | $\mathbf{57.02}_{\pm 0.61}$ | $60.90_{\pm 0.93}$ | $\mathbf{64.24}_{\pm 0.83}$ | $\underline{65.32}_{\pm 0.80}$ | $\mathbf{67.00}_{\pm 0.00}$ |
| | | WBoN | $\mathbf{52.40}_{\pm 1.39}$ | $\mathbf{57.02}_{\pm 0.61}$ | $\mathbf{61.04}_{\pm 0.88}$ | $64.20_{\pm 0.76}$ | $\mathbf{65.48}_{\pm 0.75}$ | $\mathbf{67.00}_{\pm 0.00}$ |

## E.5 ABLATION STUDY

This section presents an ablation study on our token-level uncertainty estimation method using low-rank perturbations. Appendix E.5.1 examines the effect of varying perturbation strength $\sigma_q$, while Appendix E.5.2 analyzes the impact of different decoding temperatures. In Appendix E.5.3, we investigate the effect of length normalization in the context of hallucination detection tasks. Finally, in Appendix E.5.4, we assess the validity of Assumption 3.1 as it pertains to TokUR.

Table 9: **Uncertainties for Incorrect Reasoning Path Detection.** AUROC, AUPRC, and ACC* are all reported as percentage (%), where ACC* (%) denotes the accuracy of the Top 50% generations identified by different uncertainty measures. Rows with shading indicate methods *without* **L**ength **N**ormalization (**LN**) for uncertainty estimation.

| Method | MATH500 | | | GSM8K | | | DeepScaleR | | |
|---|---|---|---|---|---|---|---|---|---|
| | AUROC | AUPRC | ACC* | AUROC | AUPRC | ACC* | AUROC | AUPRC | ACC* |
| Llama-3.2-1B-Instruct | | | | | | | | | |
| SE | 47.29±3.81 | 25.71±2.33 | 24.13±4.42 | 50.64±4.44 | 45.09±0.72 | 42.62±0.16 | 46.30±0.21 | 12.94±0.23 | 12.58±0.49 |
| SAR | 44.57±2.04 | 24.03±2.53 | 21.07±1.62 | 50.28±0.97 | 43.24±0.89 | 43.95±0.77 | 43.14±1.42 | 12.34±0.35 | 11.14±0.47 |
| $U_{Ecc}$ | 48.75±1.05 | 25.79±1.83 | 25.20±0.33 | 49.05±0.46 | 60.02±0.44 | 59.62±0.22 | 48.68±0.24 | 13.77±0.29 | 14.23±0.45 |
| $U_{Deg}$ | 60.57±2.31 | 36.32±2.59 | 30.93±0.94 | 66.60±0.36 | 75.72±0.36 | 71.99±0.39 | 56.88±0.54 | 18.04±0.63 | 16.50±0.39 |
| P(True) | 54.38±1.20 | 26.39±1.26 | 27.60±1.18 | 56.64±0.04 | 48.22±0.03 | 48.92±0.00 | 59.58±0.43 | 17.48±0.25 | 17.52±0.50 |
| LLM-Check | 56.41±0.96 | 27.01±1.22 | 31.33±1.29 | 71.01±0.02 | 61.29±0.08 | 59.54±0.00 | 55.76±0.48 | 14.55±0.26 | 17.30±0.51 |
| INSIDE | 55.71±4.69 | 28.82±4.05 | 29.20±4.33 | 53.66±0.92 | 46.03±0.23 | 45.79±1.25 | 54.73±0.82 | 15.50±0.48 | 16.30±0.35 |
| PE | 57.08±0.89 | 26.88±1.05 | 31.33±0.82 | 71.21±0.03 | 61.61±0.08 | 59.85±0.00 | 56.09±0.46 | 14.74±0.23 | 17.33±0.92 |
| LL | 55.41±0.54 | 25.88±0.87 | 29.87±0.82 | 69.01±0.03 | 58.51±0.09 | 57.38±0.00 | 53.84±0.47 | 13.93±0.23 | 16.83±0.48 |
| - LN | 79.38±0.27 | 54.64±0.75 | 43.73±0.61 | 73.67±0.00 | 68.88±0.00 | 60.92±0.00 | 82.62±0.01 | 45.76±0.02 | 25.43±0.02 |
| Self-Certainty | 71.17±0.30 | 48.37±0.50 | 38.13±0.61 | 73.41±0.00 | 68.38±0.00 | 61.38±0.00 | 71.93±0.04 | 33.81±0.08 | 21.76±0.04 |
| - LN | 23.76±0.34 | 17.04±0.10 | 11.20±0.00 | 34.42±0.00 | 34.20±0.00 | 31.54±0.00 | 21.33±0.02 | 8.57±0.01 | 4.04±0.00 |
| DeepConf | 71.77±0.12 | 46.00±0.42 | 39.87±0.46 | 75.70±0.00 | 69.72±0.00 | 62.77±0.00 | 71.65±0.04 | 29.99±0.05 | 22.00±0.04 |
| - LN | 25.79±0.34 | 17.41±0.10 | 11.47±0.23 | 38.84±0.00 | 36.22±0.00 | 35.23±0.00 | 23.87±0.01 | 8.80±0.01 | 4.91±0.02 |
| TokUR (TU, Ours) | 57.14±0.81 | 26.92±0.98 | 31.87±1.00 | 70.92±0.04 | 61.32±0.13 | 58.92±0.15 | 56.20±0.49 | 14.79±0.20 | 17.52±0.53 |
| - LN | 80.64±0.29 | 56.79±0.74 | 44.67±0.46 | 75.07±0.05 | 70.29±0.07 | 62.31±0.00 | 83.55±0.02 | 47.56±0.04 | 25.71±0.02 |
| TokUR (AU, Ours) | 56.95±0.82 | 26.81±0.99 | 31.60±0.98 | 70.90±0.05 | 61.26±0.13 | 58.87±0.32 | 56.02±0.49 | 14.73±0.19 | 17.47±0.47 |
| - LN | 80.61±0.27 | 56.73±0.75 | 44.67±0.46 | 75.03±0.06 | 70.22±0.05 | 62.21±0.18 | 83.52±0.02 | 47.48±0.05 | 25.71±0.02 |
| TokUR (EU, Ours) | 61.64±0.97 | 31.07±1.31 | 33.20±1.42 | 65.98±0.75 | 60.02±0.82 | 56.05±0.73 | 62.10±0.09 | 17.73±0.35 | 19.10±0.29 |
| - LN | 79.74±0.21 | 56.64±0.41 | 44.13±0.83 | 71.79±0.80 | 66.40±1.02 | 59.74±1.00 | 82.87±0.32 | 46.76±0.38 | 25.52±0.11 |
| Negative Length | 76.27±0.34 | 49.55±1.00 | 41.87±0.46 | 65.69±0.00 | 56.72±0.00 | 56.87±0.18 | 78.74±0.02 | 35.97±0.07 | 24.48±0.04 |
| Llama-3.1-8B-Instruct | | | | | | | | | |
| SE | 62.93±0.90 | 55.21±1.04 | 55.73±0.83 | 55.61±3.36 | 87.16±1.14 | 86.77±1.01 | 67.68±0.94 | 35.18±1.00 | 35.55±0.37 |
| SAR | 69.42±2.19 | 63.74±3.03 | 59.20±1.06 | 60.16±2.22 | 89.24±0.74 | 87.99±0.81 | 73.01±0.28 | 42.89±0.65 | 37.51±0.12 |
| $U_{Ecc}$ | 50.23±2.23 | 49.48±2.44 | 49.60±2.04 | 47.47±2.15 | 84.69±0.89 | 84.87±1.17 | 50.16±0.66 | 25.08±0.18 | 25.48±0.53 |
| $U_{Deg}$ | 58.62±0.36 | 57.69±0.90 | 53.47±1.64 | 67.22±1.06 | 92.24±0.53 | 92.62±0.88 | 59.14±0.37 | 32.64±0.43 | 29.75±0.36 |
| P(True) | 33.41±0.25 | 36.05±0.55 | 35.33±0.19 | 41.94±0.01 | 82.19±0.00 | 82.77±0.00 | 33.64±0.20 | 18.06±0.06 | 16.23±0.02 |
| LLM-Check | 57.41±0.44 | 49.69±1.07 | 52.80±1.38 | 73.98±0.01 | 93.37±0.01 | 93.23±0.00 | 55.42±0.27 | 26.46±0.19 | 28.37±0.40 |
| INSIDE | 62.94±1.72 | 55.06±3.19 | 57.33±1.01 | 58.86±2.11 | 87.44±0.94 | 88.21±0.90 | 67.05±0.49 | 33.83±0.42 | 34.13±0.10 |
| PE | 57.98±0.49 | 49.72±0.84 | 53.07±0.94 | 74.03±0.01 | 93.37±0.00 | 93.23±0.00 | 55.90±0.23 | 26.80±0.16 | 28.65±0.22 |
| LL | 55.36±0.49 | 47.24±0.90 | 51.07±0.94 | 72.21±0.02 | 92.64±0.00 | 92.46±0.00 | 52.82±0.32 | 24.48±0.13 | 26.85±0.19 |
| - LN | 81.36±0.50 | 78.80±0.36 | 72.27±0.92 | 80.03±0.01 | 95.30±0.00 | 95.08±0.00 | 84.58±0.07 | 63.92±0.03 | 43.69±0.14 |
| Self-Certainty | 76.41±0.61 | 76.22±0.87 | 69.07±0.83 | 80.60±0.11 | 95.65±0.03 | 96.26±0.09 | 76.72±0.09 | 56.15±0.30 | 39.03±0.23 |
| - LN | 21.94±0.58 | 33.57±0.42 | 28.40±1.06 | 26.44±0.01 | 75.43±0.00 | 77.85±0.00 | 18.80±0.21 | 15.15±0.10 | 8.17±0.28 |
| DeepConf | 71.86±0.70 | 69.57±0.94 | 66.27±1.15 | 83.30±0.07 | 96.23±0.00 | 96.56±0.09 | 73.05±0.08 | 48.76±0.10 | 37.48±0.14 |
| - LN | 23.73±0.62 | 34.08±0.44 | 30.80±1.06 | 30.86±0.01 | 77.30±0.00 | 79.38±0.00 | 21.20±0.24 | 15.50±0.11 | 9.27±0.16 |
| TokUR (TU, Ours) | 56.49±0.46 | 48.24±0.85 | 52.13±0.75 | 73.98±0.05 | 93.27±0.04 | 93.13±0.09 | 54.86±0.17 | 25.97±0.12 | 27.97±0.17 |
| - LN | 82.47±0.47 | 79.62±0.33 | 74.00±0.69 | 81.01±0.04 | 95.53±0.05 | 95.54±0.00 | 85.33±0.07 | 65.25±0.01 | 43.91±0.09 |
| TokUR (AU, Ours) | 56.31±0.47 | 48.11±0.84 | 51.87±0.68 | 73.97±0.02 | 93.26±0.03 | 93.13±0.09 | 54.77±0.18 | 25.90±0.13 | 27.93±0.11 |
| - LN | 82.43±0.48 | 79.56±0.35 | 74.00±0.69 | 80.97±0.02 | 95.52±0.03 | 95.49±0.00 | 85.31±0.07 | 65.20±0.02 | 43.89±0.08 |
| TokUR (EU, Ours) | 60.92±0.46 | 52.64±0.71 | 56.13±1.36 | 67.92±0.72 | 92.41±0.24 | 92.15±0.41 | 57.42±0.23 | 28.32±0.16 | 29.65±0.10 |
| - LN | 82.86±0.42 | 81.35±0.66 | 72.40±1.20 | 78.31±1.58 | 94.91±0.59 | 94.67±0.77 | 84.92±0.28 | 65.57±0.43 | 43.89±0.27 |
| Negative Length | 78.11±0.54 | 73.81±0.28 | 68.80±0.40 | 73.64±0.01 | 93.36±0.00 | 93.54±0.00 | 81.20±0.20 | 57.12±0.31 | 41.57±0.16 |

### E.5.1 THE EFFECT OF PERTURBATION STRENGTH $\sigma_q$ ON UNCERTAINTY ESTIMATION

To investigate the impact of perturbation strength on uncertainty estimation, **we conducted a series of experiments under varying $\sigma_q$ settings, as shown in Fig. 4.** First, we computed the average uncertainty estimates (TU, AU, and EU) on samples generated from the GSM8K test dataset using Llama-3.2-1B-Instruct. As illustrated in Fig. 4 **Left**, the model's uncertainty increases steadily with higher perturbation strength. However, once $\sigma_q$ exceeds a critical threshold (e.g., 0.2), a sharp rise in uncertainty is observed. This rise illustrates that the current approximate distribution of the weights has deviated too far from the pre-trained point estimation of the parameters, leading to unreliable uncertainty estimates.

**We further evaluate the effect of perturbation strength on downstream task performance.** Specifically, we assess how effectively the uncertainty estimates, obtained under different $\sigma_q$ values,

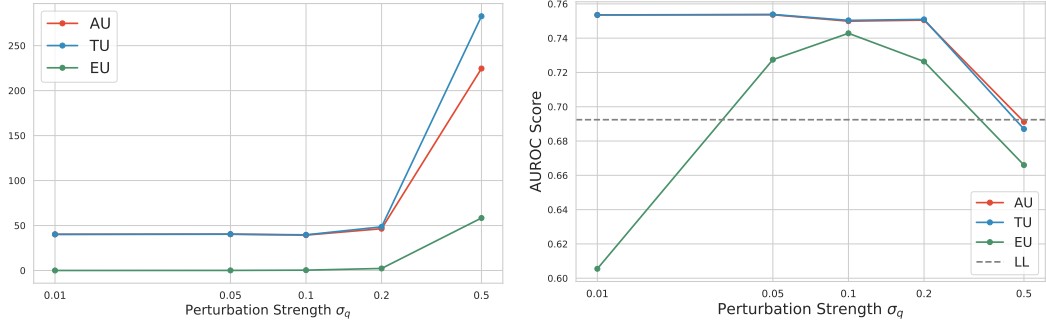

Figure 4: **Left:** Uncertainty estimation with different perturbation strength $\sigma_q$. **Right:** Influence of perturbation strength on uncertainty-based AUROC scores.

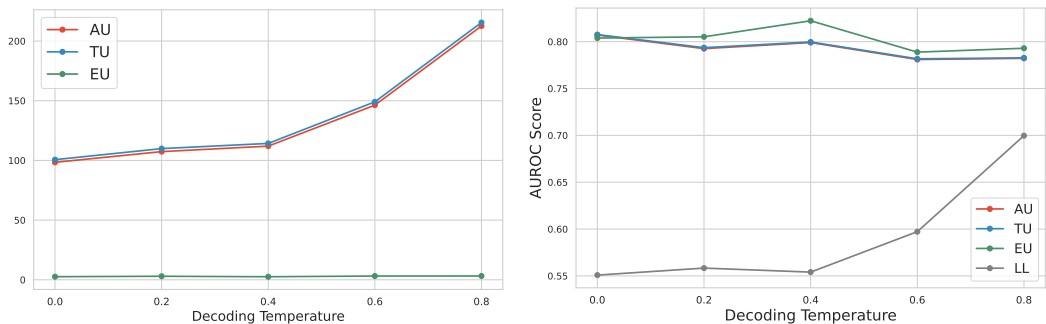

Figure 5: **Left:** Uncertainty estimations in different token decoding temperature $\tau$. **Right:** Influence of token decoding temperature on uncertainty-based AUROC scores.

can be used as scoring signals to distinguish between correct and incorrect samples, as described in Sec. 4.1.2. As shown in Fig. 4 **Right**, for `TokUR` (EU), too small $\sigma_q$ does not lead to meaningful improvements in log-likelihood, whereas an excessively large $\sigma_q$ harms performance of all three `TokUR` variants (AU, TU and EU) by distorting the original semantic content. Based on these findings, we set $\sigma_q = 0.1$ for the experiments reported in Sec. 4.

### E.5.2  THE EFFECT OF TOKEN DECODING TEMPERATURE $\tau$ ON UNCERTAINTY ESTIMATION

During text generation with large language models, the decoding temperature introduces uncertainty into the model's output. In general, higher temperatures lead to more diverse responses. In this section, we investigate the relationship between decoding temperature $\tau$ and uncertainties estimated by our token-level approach. Specifically, we use `Llama-3.2-1B-Instruct` to answer questions from the MATH500 dataset under different decoding temperature settings and estimate the average uncertainty of the model's responses.

As shown in Fig. 5 **Left**, increasing the decoding temperature $\tau$ results in a notable rise in token-level Aleatoric Uncertainty (AU) of the model, whereas the Epistemic Uncertainty (EU) remains relatively unaffected. Additionally, we report the AUROC scores of various uncertainty estimation approaches across different temperature settings in Fig. 5 **Right**. These results indicate that varying the temperature $\tau$ does not harm the performance of `TokUR`, highlighting its robustness to changes in decoding temperature.

### E.5.3  ABLATION STUDY OF LENGTH NORMALIZATION

Length normalization is a standard technique for aggregating *token-level* uncertainty into *sequence-level* uncertainty (Fu et al., 2025; Kang et al., 2025), as it mitigates the bias introduced by sequence length when evaluating generation confidence. However, as described in Eqn. 13, we do not apply

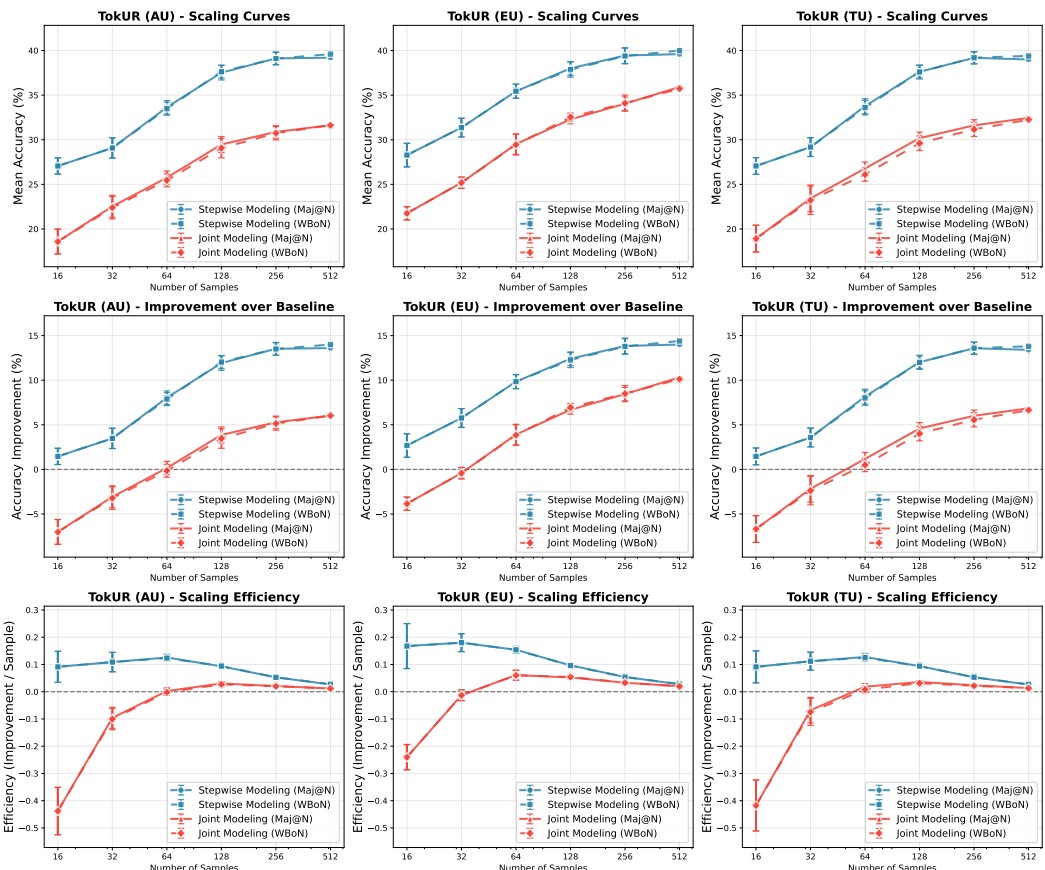

Figure 6: **Ablation of stepwise posterior sampling.** Comparison of *stepwise* vs. *joint* modeling on `Llama-3.2-1B-Instruct` across accuracy, improvement, and efficiency. Stepwise modeling consistently achieves better scaling performance, validating Assumption 3.1.

normalization when computing `TokUR`. To assess the impact of sequence length on uncertainty estimation, we therefore conduct an ablation study on length normalization.

**Experimental Setup.**  We investigate the effect of length normalization on incorrect reasoning path detection across three datasets (MATH500, GSM8K, and DEEPSCALER), following the same settings as in Table 1. We compare `TokUR` with and without **L**ength **N**ormalization (**LN**), along with representative baselines. In addition, we introduce a naive baseline, **Negative Length**, which uses sequence length alone as a confidence signal.

**Results.**  As shown in Table 9, the impact of length normalization varies significantly across methods. For both **LL** and `TokUR`, normalization consistently reduces AUROC and AUPRC, indicating that raw sequence length introduces a favorable bias that benefits uncertainty aggregation in de-hallucination tasks. This observation is further reinforced by the strong performance of the **Negative Length** baseline, which alone achieves competitive results across all datasets. In contrast, Self-Certainty and DeepConf show clear gains with normalization (e.g., Self-Certainty improves from 23.76 to 71.17 AUROC on MATH500), suggesting that normalization is essential for stabilizing their performance. Overall, these findings reveal that the role of length normalization is highly method-dependent.

### E.5.4   ABLATION STUDY OF STEPWISE POSTERIOR SAMPLING

To examine the validity of Assumption 3.1, **we perform an ablation study comparing *step-wise* posterior sampling against the *joint* posterior formulation.** Concretely, we evaluate

test-time scaling on MATH500 dataset, using `TokUR` with both stepwise and joint modeling on `Llama-3.2-1B-Instruct`, while keeping all other settings consistent with Table 4.

As shown in Fig. 6, stepwise modeling consistently outperforms joint modeling across all uncertainty measures (AU, EU, and TU). Specifically, stepwise sampling achieves higher mean accuracy and larger improvements over the baseline, while also demonstrating superior scaling efficiency with increasing numbers of samples. These results provide strong empirical support for our assumption that posterior samples should not be shared across decoding steps, validating the design choice in Assumption 3.1.

### E.6 CASE STUDY

In this section, we present several representative examples from the MATH500 and GSM8K datasets, along with their corresponding solutions generated by `Llama-3.2-1B-Instruct`. We estimate token-level uncertainty for each output using the definitions provided in Eqn. 10~Eqn. 12. The visualizations are shown in Fig. 7~Fig. 10, where Aleatoric Uncertainty (AU, in RED) and Epistemic Uncertainty (EU, in GREEN) are visualized as text-heatmap. The background shading of each token corresponds to the magnitude of its uncertainty: the darker the shade, the higher the uncertainty, indicating a lower model confidence for that token.

We observe that incorrect solutions consistently exhibit elevated uncertainty at or near the token where the wrong final answer is generated. For instance, as shown in Fig. 7, sharp spikes in uncertainties happens with the arithmetic error of reversing "9600 - 7200" into "7200 - 9600". In contrast, correct solutions tend to show lower uncertainty overall and maintain low uncertainty on key answer tokens.

Furthermore, incorrect outputs tend to contain a higher density of high-uncertainty tokens throughout the solution, whereas correct outputs are generally more consistent and confident. These observations suggest that our token-level uncertainty estimation method can serve as a useful signal for identifying potential reasoning failures or unreliable outputs, offering a valuable diagnostic tool for both model interpretability and downstream error detection.

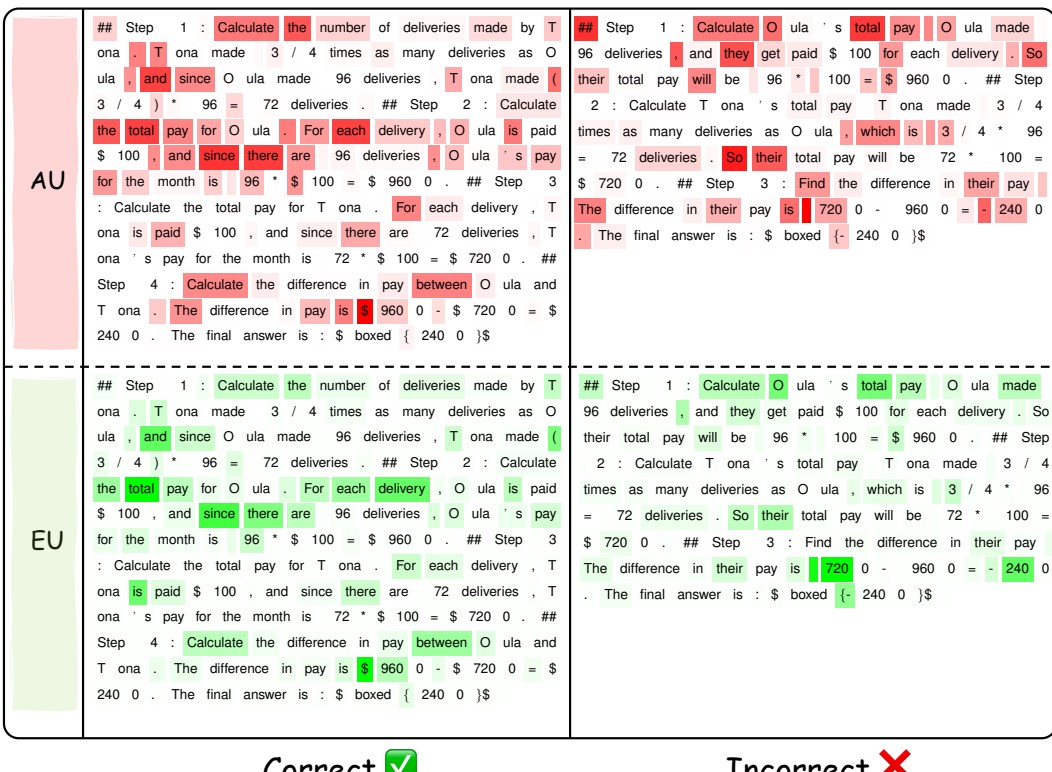

Figure 7: **Case Study (1/4):** The sample is from GSM8K, whose correct answer is 2400. In the incorrect solution, the model demonstrated significant uncertainty when mistakenly reversing "9600 − 7200" as "7200 − 9600", and also exhibited high uncertainty at the negative sign "−" in the final answer.

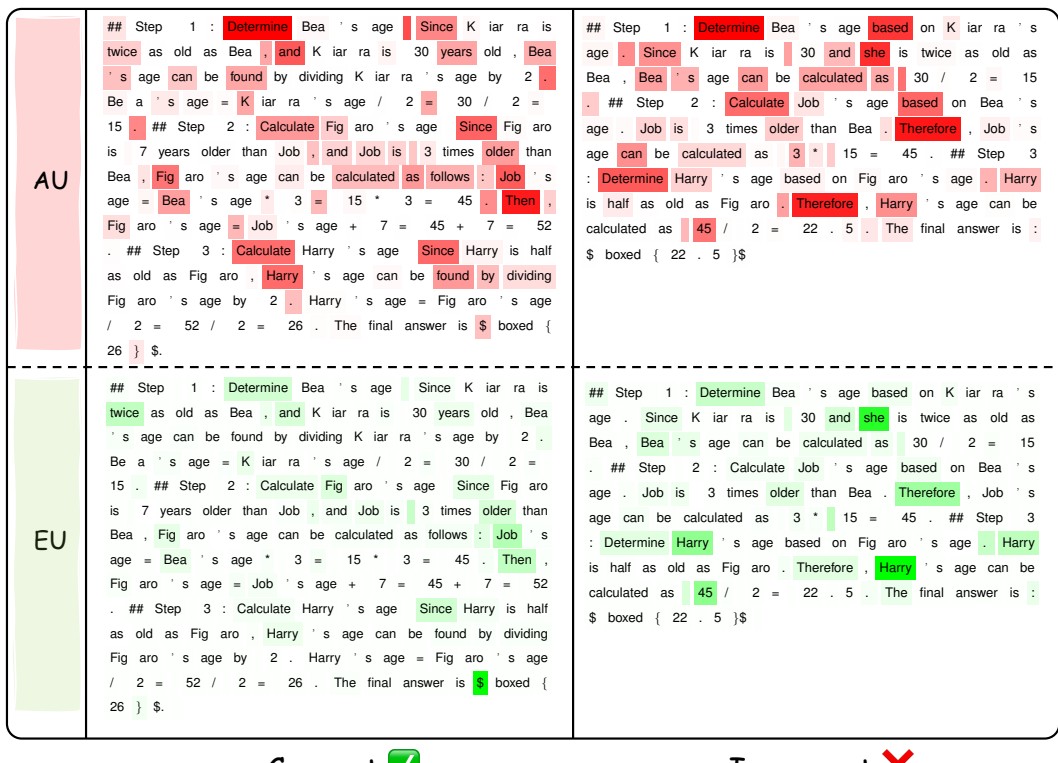

Figure 8: **Case Study (2/4):** The sample is from GSM8K. In this example, the incorrect solution ignores the critical condition that "Figaro is 7 years older than Job," leading to the use of 45 instead of 52 in the final calculation. Notably, the model exhibits high uncertainty at the token "45" indicating a lack of confidence in its own response at that point.

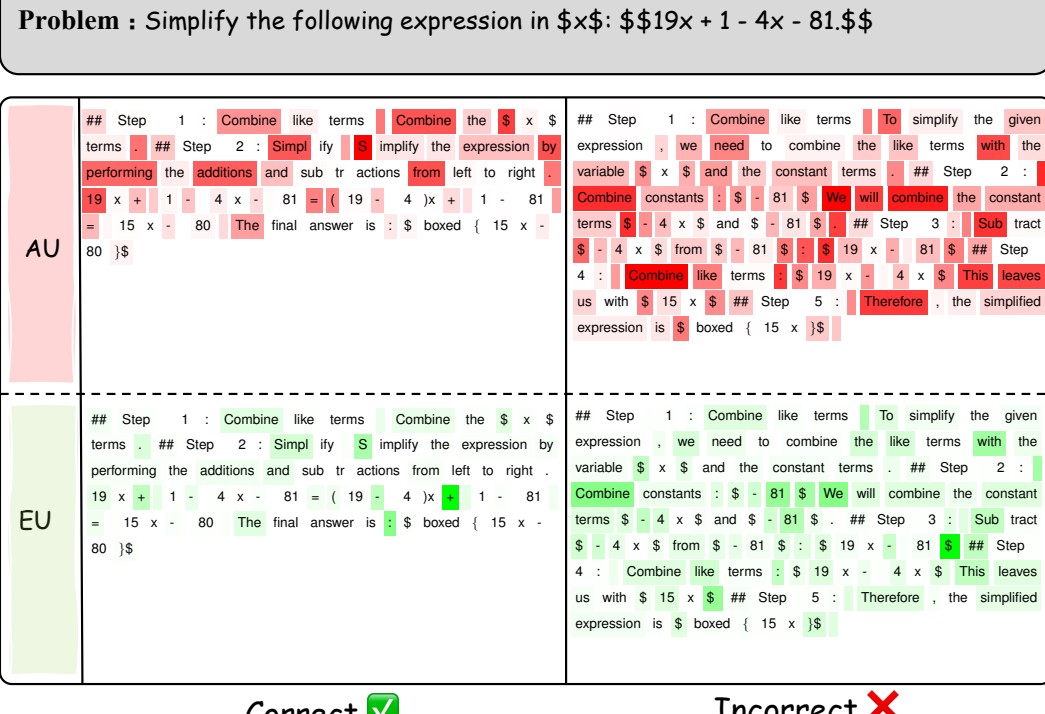

Figure 9: **Case Study (3/4):** The sample is from MATH500. In this example, the incorrect solution gives its final answer "$15x$" in step 4. The model exhibits high uncertainty at the token next to "$15x$" because it overlooks the constant term. Furthermore, it can be observed that tokens associated with high uncertainty occur more frequently in the incorrect solution.

**Problem :** In regular pentagon $FGHIJ$, extending the sides of the pentagon, as shown, forms a star. What is the measure of angle $A$ in the figure?\n\n [IMAGE]

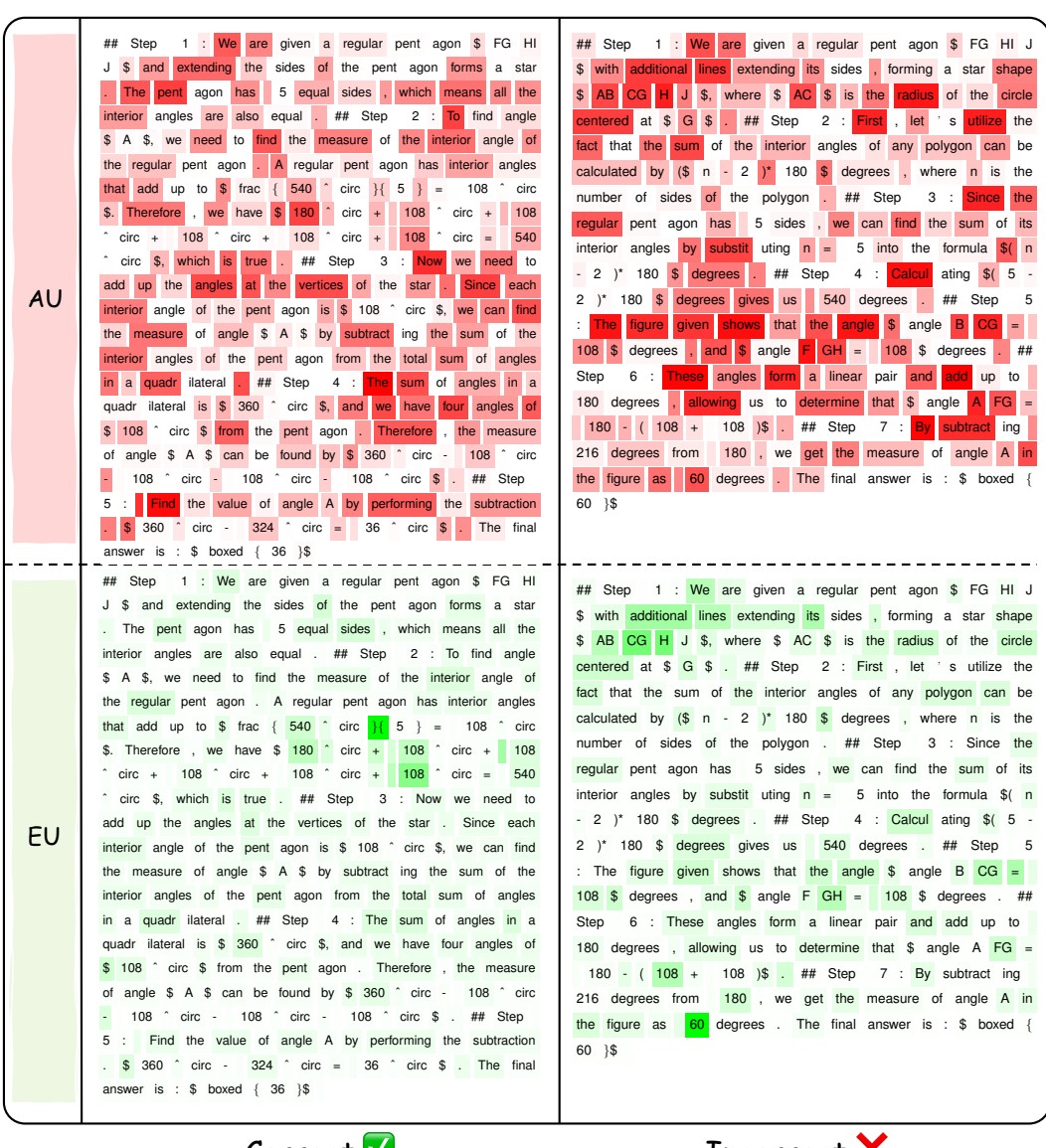

Correct ✅          Incorrect ❌

Figure 10: **Case Study (4/4):** The sample is from MATH500. In this example, the model demonstrated notably high uncertainty at the incorrect answer token "60". In the correct solution on the left, the model had low uncertainty for the correct answer "36".

