# OpenReview forum: "TokUR: Token-Level Uncertainty Estimation for Large Language Model Reasoning"
_ICLR.cc/2026/Conference — ICLR 2026 Poster_

### Official Review · Reviewer_YhsK · 2025-10-31

**Soundness:** 2
**Presentation:** 2
**Contribution:** 3
**Rating:** 4
**Confidence:** 3

**Summary:**

This paper proposes the TokUR method, which injects low-rank weights to approximate the weight posterior distribution in Bayesian modeling of LLMs. During the decoding stage, it captures uncertainty in the model, leveraging quantized uncertainty to simultaneously estimate the semantic uncertainty of generated responses. Experimental results demonstrate that the proposed method produces uncertainty estimates that correlate well with task difficulty, and further show its effectiveness in detecting erroneous reasoning paths and improving answer quality.

**Strengths:**

1. This paper introduces a novel approach for estimating uncertainty in long-form reasoning and demonstrates its effectiveness.
2. The paper is clearly written and easy to follow.
3. The experimental design is fairly comprehensive.

**Weaknesses:**

1. The authors decompose uncertainty into Aleatoric Uncertainty (AU) and Epistemic Uncertainty (EU). However, the definition of AU appears to correspond more to uncertainty in the input, while EU is more related to uncertainty in the model output. Since EU focuses on the model’s prediction behavior, it should theoretically better reflect the model’s capability issues. Nevertheless, the experimental results do not support this, and the paper does not provide analysis or interpretation of the three uncertainty estimates. I believe such analysis should be included.

2. The expressions for Query-Level Uncertainty and Response-Level Uncertainty both involve (y), and the definitions appear highly similar, which may cause confusion or misunderstanding. I suggest the authors explicitly clarify the difference and explain the rationale behind these definitions to avoid ambiguity for readers.

3. Assumption 3.1 ignores temporal correlations among parameters. The impact of this approximation needs to be justified either qualitatively or quantitatively to demonstrate the validity of the assumption.

4. Some important related work is missing in the discussion. Many existing studies on uncertainty estimation and LLM reasoning theory are not referenced. For example:
    *  *Position: Uncertainty Quantification Needs Reassessment for Large Language Model Agents* (ICML 2025), which discusses the relevance of uncertainty estimation.
    * *A Theoretical Study on Bridging Internal Probability and Self-Consistency for LLM Reasoning* (NeurIPS 2025), which provides theoretical insights into reasoning errors in LLMs to improve reasoning accuracy.
    * *From Calibration to Collaboration: LLM Uncertainty Quantification Should Be More Human-Centered* (arXiv:2506.07461), which analyzes the limitations of existing LLM uncertainty estimation approaches.

**Questions:**

See weaknesses above.

---

> ### Author Response · Authors · 2025-11-21
> **Official Rebuttal by Authors (1/2)**
>
> Thank you for your constructive feedback. We are glad that you find our work `novel`, the paper `clearly written`, and our experiments `comprehensive`. Below we address your questions in detail. We will also revise the paper accordingly, with all changes marked in blue.
>
> **W1. "The authors decompose uncertainty into Aleatoric Uncertainty (AU) and Epistemic Uncertainty (EU). However, the definition of AU appears to correspond more to uncertainty in the input, while EU is more related to uncertainty in the model output. Since EU focuses on the model’s prediction behavior, it should theoretically better reflect the model’s capability issues. Nevertheless, the experimental results do not support this, and the paper does not provide analysis or interpretation of the three uncertainty estimates. I believe such analysis should be included."**
>
> Thank you for your valuable feedback.
>
> **What do EU and AU mean in our TokUR?**
> We agree that, under the classical definitions of Epistemic Uncertainty (EU) and Aleatoric Uncertainty (AU), **EU** is typically associated with uncertainty in the *model parameters*, while **AU** captures uncertainty of *inputs*, as manifested through the model’s uncertain *outputs*.
> In our work, we adapt these notions to the *token level* (`Section 3.1`), computing both types of uncertainty at each token position. Concretely, for a response token $y_t$,
> + its **AU** captures the uncertainty conditioned on the input $x$ and the previously generated tokens $y_{<t}$, while
> + its **EU** measures how sensitive the model's prediction at step $t$ is to perturbations in the model parameters.
>
> We then aggregate these token-level uncertainties to obtain *response-level* estimates as in `Definition 3.1`. Under this formulation, all three quantities (AU, EU, and TU) represent uncertainty of the final response, each highlighting a different but complementary aspect.
>
>
> **Why does EU not consistently outperform AU in TokUR?**
> As noted in prior works [1, 3-5], **aleatoric uncertainty captures noise inherent in the observations or latent variables**, while **epistemic uncertainty arises from uncertainty in the model parameters**. These two forms of uncertainty reflect *distinct sources of uncertainties* and need not behave consistently across all tasks or model families.
> In practice, due to differences in *model size, training corpus, architectural inductive biases, and task difficulty,* **it is not guaranteed that EU should always dominate AU, or AU should be better than EU**. Indeed, prior studies have repeatedly reported that **AU is sometimes the stronger signal, while in other settings EU becomes more informative** [1-3, 6], indicating that their relative strengths vary across tasks and model families.
>
>
> **W2. "The expressions for Query-Level Uncertainty and Response-Level Uncertainty both involve (y), and the definitions appear highly similar, which may cause confusion or misunderstanding. I suggest the authors explicitly clarify the difference and explain the rationale behind these definitions to avoid ambiguity for readers."**
>
> This is a good point, and we are sorry for any confusion.
> + In `Definition 2.1 (Query-Level Uncertainty)`, **$\mathbf{y}$ is a random variable.** In the Query-Level Uncertainty $U(\mathbf{y}|x)$, the symbol $\mathbf{y}$ represents the **entire random response space** of the predictive distribution $p(\mathbf{y}|x)$. It means we consider *all possible responses* the model may produce for input $x$. Here the uncertainty is computed by taking expectations over **the full predictive distribution**, i.e., integrating over all possible $y$. Thus, Query-Level Uncertainty measures **how inconsistent (uncertain) the model is about answering the query $x$**.
> + In `Definition 3.1 (Response-Level Uncertainty)`, **$y$ is \*not\* a random variable, but is a concrete generated sample.** In Response-Level Uncertainty, we define
> $$
> \tilde{U}(y|x) = \sum\_{t=1}^T U(\mathbf{y}\_t \mid y\_{<t}, x).
> $$
> Here, the $y$ is **a specific generated response**, not a random variable. We compute this uncertainty based on a *particular sampled output sequence* $y = (y_1,\dots,y_T)$. The Response-Level Uncertainty is computed token by token along **this specific trajectory**, not over all possible trajectories. Thus, Response-Level Uncertainty measures **how uncertain the model was while producing this particular response**.
>
> We appreciate the reviewer’s suggestion and will clarify this distinction explicitly in the revised paper to improve readability.

---

> > ### Author Response · Authors · 2025-11-21
> > **Official Rebuttal by Authors (2/2)**
> >
> > **W3. "Assumption 3.1 ignores temporal correlations among parameters. The impact of this approximation needs to be justified either qualitatively or quantitatively to demonstrate the validity of the assumption."**
> >
> > We are sorry for the confusion.
> > As noted in `Section 3.1`, when introducing `Assumption 3.1`, we explicitly state that stepwise posterior sampling is only one of many valid probabilistic formulations. Compared to joint modeling, the assumption adopted in TokUR aligns more naturally with the autoregressive structure of LLMs, making it a practical and compatible choice.
> >
> >
> > Empirically, to assess this assumption, we conducted a dedicated ablation study comparing two posterior formulations: **stepwise posterior sampling** (our `Assumption 3.1`) versus the **joint posterior formulation** (i.e., with temporal correlation that you mentioned). The detailed quantitative results are provided in `Appendix E.4.4`.
> > As shown in `Figure 6`, stepwise modeling **consistently outperforms** joint modeling across all uncertainty metrics (AU, EU, and TU), and across *accuracy, improvement, and scaling-efficiency curves,* including both mean performance and scaling trends as the number of samples increases.
> > These results provide strong empirical evidence that treating posterior samples independently across decoding steps leads to **better** uncertainty estimation in practice. Therefore, the ablation directly supports the soundness of `Assumption 3.1` and validates the design choice adopted in TokUR.
> >
> > **W4. "Some important related work is missing in the discussion. Many existing studies on uncertainty estimation and LLM reasoning theory are not referenced."**
> >
> > Thank you for pointing us to these interesting papers. We appreciate the your suggestions regarding additional literature on uncertainty estimation. We will incorporate all of the recommended references in the next revision and update `Section 5 (Related Work)` to more clearly position our TokUR within the broader context.
> >
> >
> >
> > **References**
> > + [1] Kendall A, Gal Y. What uncertainties do we need in bayesian deep learning for computer vision?. Advances in neural information processing systems, 2017
> > + [2] Valdenegro-Toro et al. A deeper look into aleatoric and epistemic uncertainty disentanglement. CVPR 2022
> > + [3] Depeweg S et al. Uncertainty decomposition in bayesian neural networks with latent variables. arXiv preprint 2017
> > + [4] Der et al. Aleatory or epistemic? Does it matter?. Structural safety 2009
> > + [5] Gal Y. Uncertainty in deep learning. 2016.
> > + [6] Malinin A et al. Predictive uncertainty estimation via prior networks. Advances in neural information processing systems, 2018

---

### Official Review · Reviewer_ZNDU · 2025-11-01

**Soundness:** 3
**Presentation:** 3
**Contribution:** 2
**Rating:** 6
**Confidence:** 3

**Summary:**

This paper introduces the TokUR framework for uncertainty quantification in large language models (LLMs), targeting complex, multi-step mathematical reasoning tasks. TokUR estimates uncertainty at the token level by introducing low-rank random weight perturbations during decoding, enabling principled decompositions into aleatoric and epistemic uncertainty for each token and entire responses.

**Strengths:**

- Most technical concepts are well-explained and visualized, aiding algorithmic understanding.

- The approach itself is elegant.

- It addresses a key practical need for safe and reliable LLM deployment, especially where confidence calibration is critical.

**Weaknesses:**

- Experiments focus on mathematical reasoning; application to verbal reasoning, open-ended tasks, or black-box LLMs is not shown.

- Repeated sampling during inference, while compatible with vLLM, is still non-trivial and could limit deployment in latency-sensitive use cases.

**Questions:**

- How does TokUR perform on open-domain, verbal, or dialogue tasks?

- How much cost does in incur to get perturbed weights?

- How do these perturbed weights affect the performance of the model?

---

> ### Author Response · Authors · 2025-11-21
> **Official Rebuttal by Authors (1/2)**
>
> Thank you for your constructive feedback. We appreciate your positive assessment that our technical concepts are `well-explained and visualized` and that our approach is `elegant`. We are also glad that you find our work `addresses a key practical need for safe and reliable LLM deployment`. Below we address your question in detail. We will also revise the paper accordingly, with all changes marked in blue.
>
> **W1&Q1. "Experiments focus on mathematical reasoning; application to verbal reasoning, open-ended tasks, or black-box LLMs is not shown."**
>
> This is a good point. **Our TokUR is designed to estimate uncertainty in long-form generation**, an area that remains largely unexplored in uncertainty estimation research (as stated in `Section 4 (Experiments)`). We use mathematical reasoning as our primary evaluation domain because it is both practically important and representative: it (i) requires long-form, chain-of-thought reasoning traces and (ii) provides verifiable ground-truth answers that enable objective assessment.
>
> Following your suggestion, we further evaluate the generalizability of TokUR beyond mathematical reasoning by conducting additional experiments on **five non-math long-form generation tasks across multiple domains,** including:
> + **Logical Reasoning:** We evaluate three tasks from **Reasoning Gym** [1]: Zebra Puzzles, Leg Counting, and Color-Cube Rotation. For each task, we sample 100 instances and prompt the model using Chain-of-Thought (CoT). We report AUROC, AUPRC, and Top-50% ACC following the setting of `Section 4.1.1` in our paper.
> + **Code Generation:** We evaluate on **HumanEval** [2], a standard benchmark for code generation. We report AUROC, AUPRC, and Top-50% ACC.
> + **Truthfulness:** We evaluate on **FactScore** [3], a widely used benchmark for factual correctness. We use *GPT-5-mini* to decompose model outputs into atomic facts and judge them. Here we report Top-50% Precision (FactScore) when ranking responses by uncertainty.
>
> Given the increased difficulty of the newly added tasks and the model-diversity concern raised in `W3`, we perform all additional evaluations using a different base LLM, *Qwen-2.5-3B-Instruct.* The results are summarized in Table A.
>
> **Table A. Performance of Uncertainty Estimation Methods for Incorrect Reasoning Detection on 5 Long-Form Generation Tasks with Qwen-2.5-3B-Instruct.**
>
> | Method | Zebra Puzzles | | | Leg Counting | | | Color Cube | | | HumanEval | | | FactScore |
> | - | - | - | - | - | - | - | - | - | - | - | - | - | - |
> | | AUROC | AUPRC | ACC | AUROC | AUPRC | ACC | AUROC | AUPRC          | ACC            | AUROC          | AUPRC          | ACC            | Precision      |
> | PE               | 44.70±0.00     | 21.76±0.00     | 24.00±0.00     | 38.87±0.25     | 27.44±0.05     | 26.00±0.00     | 46.73±0.00     | 28.75±0.00     | 22.00±0.00     | 60.52±0.00     | 81.45±0.00     | 78.05±0.00     | 41.86±0.02     |
> | LL               | 42.15±0.00     | 21.02±0.00     | 22.00±0.00     | 37.92±0.43     | 27.10±0.13     | 32.67±1.15     | 46.36±0.00     | 26.01±0.00     | 24.00±0.00     | 56.42±0.00     | 79.48±0.00     | 75.61±0.00     | 45.03±0.38     |
> | Self-Certainty   | 47.77±0.00     | 22.95±0.00     | 30.00±0.00     | 50.46±0.30     | 32.88±0.37     | 34.00±0.00     | 50.62±0.00     | 30.80±0.00     | 28.00±0.00     | 61.48±0.00     | 80.27±0.00     | 80.49±0.00     | 40.10±0.38     |
> | DeepConf         | 42.41±0.00     | 21.07±0.00     | 26.00±0.00     | 42.48±0.71     | 28.96±0.32     | 30.00±0.00     | 49.01±0.00     | 24.41±0.00     | 24.00±0.00     | 63.60±0.00     | **82.75±0.00** | 80.49±0.00     | 41.96±0.49     |
> | TokUR (TU, Ours) | 71.38±0.29     | 41.42±0.38     | **39.33±1.15** | 69.58±2.05     | 48.16±0.99     | 48.67±1.15     | 58.33±2.86     | 31.14±1.56   | 31.33±1.15   | 63.15±0.82     | 81.59±0.41     | **80.90±0.70** | 60.35±0.74     |
> | TokUR (AU, Ours) | **71.66±0.35** | **41.71±0.44** | **39.33±1.15** | 69.51±1.82     | 47.69±0.70     | 48.67±1.15     | 56.57±2.88     | 30.02±1.36     | 28.00±2.00     | 60.50±0.80     | 81.75±0.41     | **80.90±1.41** | **61.53±0.58** |
> | TokUR (EU, Ours) | 71.28±0.38     | 41.20±0.76     | 37.33±2.31     | **69.66±2.51** | **51.27±2.07** | **50.67±2.31** | **60.08±1.98** | **33.71±1.01** | **32.00±2.00** | **60.64±0.77** | 80.75±0.48     | 80.49±0.00     | 45.26±0.54     |
>
>
> Across all non-math tasks, **our TokUR consistently ranks as the best or second-best method, confirming that it generalizes effectively to diverse long-form generation settings and retains strong performance beyond the mathematical reasoning domain.**
>
> We will include the discussion and results above in the revision as suggested.

---

> > ### Author Response · Authors · 2025-11-21
> > **Official Rebuttal by Authors (2/2)**
> >
> > **W2&Q2. "Repeated sampling during inference, while compatible with vLLM, is still non-trivial and could limit deployment in latency-sensitive use cases. How much cost does in incur to get perturbed weights?"**
> >
> > Thank you for mentioning this. As a Bayesian approach that relies on Monte Carlo sampling for expectation estimation, **TokUR incurs approximately $N\times$** the computational cost of standard single-sample LLM inference, where $N$ denotes the number of stochastic forward passes at test time. This overhead is inherent to **all Bayesian Model Averaging (BMA) approaches**, and improving test-time efficiency is **NOT** the primary focus of this work.
> >
> > Fortunately, as discussed in `Section 3.3`, TokUR's design incorporates `a low-rank noise matrix` to minimize the additional sampling cost. **Empirically, we find that TokUR delivers strong uncertainty-estimation performance with a small sampling budget (e.g., $N=2$ to $N=5$)**. Moreover, **our implementation is fully compatible with vLLM,** enabling efficient batched sampling and low-overhead deployment.
> >
> > Inspired by your comment and for completeness, Table B below summarizes the inference cost of all query-level uncertainty estimation for both our TokUR and the baselines adopted in our experiments.
> >
> > **Table B. Inference Cost of Query-Level Uncertainty Estimation for TokUR and Baselines.**
> >
> > | Method | \#Forward Passes per Query |
> > | ------ | ------------------------------------------------------------ |
> > | SE     | $\times (N + M)$, where $M$ is the cost of an additional NLI model |
> > | SAR    | $\times (N + M)$, where $M$ is the cost of an additional semantic-similarity model |
> > | INSIDE | $\times N$ |
> > | **TokUR (Ours)** | $\times N$ |
> >
> > Note that these baselines (e.g., SE and SAR) require generating multiple *full* responses per query, often with additional auxiliary models, leading to **equal or higher total cost than TokUR**. Thus, although TokUR inherits the sampling overhead typical of Bayesian approaches, its overall cost remains competitive compared to existing query-level uncertainty estimation techniques.
> >
> > We will include the discussion above and revise the paper accordingly as suggested.
> >
> >
> > **Q3. "How do these perturbed weights affect the performance of the model?"**
> >
> > Thank you for mentioning this. In fact, TokUR is purely an uncertainty estimation framework and **does not alter the generation process of the base model**; therefore it **does not affect the performance**. As mentioned in `Section 3.3`, we always use the **mean weights** $\mathbf{W_0}$, which is the original model parameters, for decoding at every step, rather than performing BMA during generation. Consequently, perturbed weights have **no effect on the model’s generated responses.** As noted in `Section 3.3`, `This approach allows for a controlled study of the effects of uncertainty estimation itself,` and we view this clean separation between generation and uncertainty estimation as a key advantage of TokUR.
> >
> >
> > **References**
> > + [1] Stojanovski, Zafir, Oliver Stanley, Joe Sharratt, Richard Jones, Abdulhakeem Adefioye, Jean Kaddour, and Andreas Köpf. "REASONING GYM: Reasoning Environments for Reinforcement Learning with Verifiable Rewards." arXiv preprint arXiv:2505.24760 (2025).
> > + [2] Chen, Mark. "Evaluating large language models trained on code." arXiv preprint arXiv:2107.03374 (2021).
> > + [3] Min, Sewon, Kalpesh Krishna, Xinxi Lyu, Mike Lewis, Wen-tau Yih, Pang Koh, Mohit Iyyer, Luke Zettlemoyer, and Hannaneh Hajishirzi. "Factscore: Fine-grained atomic evaluation of factual precision in long form text generation." In Proceedings of the 2023 Conference on Empirical Methods in Natural Language Processing, pp. 12076-12100. 2023.

---

### Official Review · Reviewer_9t1g · 2025-11-01

**Soundness:** 3
**Presentation:** 3
**Contribution:** 2
**Rating:** 4
**Confidence:** 3

**Summary:**

The paper introduces TokUR, a framework that lets LLMs self-assess and self-improve on multi-step mathematical reasoning by estimating token-level uncertainty during decoding. TokUR applies low-rank random weight perturbations, producing predictive distributions for each generated token. These per-token signals are decomposed into aleatoric and epistemic uncertainty and then aggregated to the sequence level to capture a response’s semantic confidence. Compared to prior query-level methods and response-level methods, TokUR offers a more fine-grained and accurate uncertainty estiomation. Experiments show that TokUR’s uncertainty is strongly correlated with correctness and robustness, flags incorrect reasoning paths, selects higher-quality solutions among candidates.

**Strengths:**

1. Strong theoretical ground.

The paper’s theoretical backgrounds appear solid and thoughtfully developed. Assumptions are stated clearly.

2. Clear positioning vs. prior work.

The manuscript does an excellent job for introducing the limitations of prior methods (query-level and response-level approaches), making it easy to understand the gap this work fills.

3. Readable and well organized.

The paper is clearly written and easy to follow, with clear notations.

**Weaknesses:**

1. On task scope

It seems that the framework is “not limited to mathematical reasoning”. Is there a reason that the paper should focus especially on mathematical reasoning? A few non-math settings would support the generalizability of the framework.

2. On cost analysis

While the cost is the common challenge of uncertainty estimation, it seems that there is no thorough cost analysis in the paper, compared to prior methods like query-level uncertainty estimation. Also, how does the estimation technique used in Section 3.3 help reduce cost?

3. On model diversity

Evaluating two models from the same Llama family seems insufficient to support the correlation between accuracy and token-level uncertainty estimation. Including a few more models from different family and sizes is encouraged to reduce bias in experiments and to strengthen the claim.

**Questions:**

1. It would be helpful to analyze the length-robustness of your token-level uncertainty estimates. As responses grow longer, how do the estimation errors (or bounds) evolve?

---

> ### Author Response · Authors · 2025-11-21
> **Official Rebuttal by Authors (1/4)**
>
> Thank you for your constructive feedback. We sincerely appreciate your positive assessment that our work has a `strong theoretical ground` and a `clear positioning vs. prior work`. We are also glad that you found our discussion of prior work's limitations to be `excellent` and that our paper `clearly written and easy to follow`. Below, we address your questions in detail. We will also revise the paper accordingly, with all changes marked in blue.
>
> **W1. [Task Scope] "It seems that the framework is 'not limited to mathematical reasoning'. Is there a reason that the paper should focus especially on mathematical reasoning? A few non-math settings would support the generalizability of the framework."**
>
> This is a good point. **Our TokUR is designed to estimate uncertainty in long-form generation**, an area that remains largely unexplored in uncertainty estimation research (as stated in `Section 4 (Experiments)`). We use mathematical reasoning as our primary evaluation domain because it is both practically important and representative: it (i) requires long-form, chain-of-thought reasoning traces and (ii) provides verifiable ground-truth answers that enable objective assessment.
>
> Following your suggestion, we further evaluate the generalizability of TokUR beyond mathematical reasoning by conducting additional experiments on **five non-math long-form generation tasks across multiple domains,** including:
> + **Logical Reasoning:** We evaluate three tasks from **Reasoning Gym** [1]: Zebra Puzzles, Leg Counting, and Color-Cube Rotation. For each task, we sample 100 instances and prompt the model using Chain-of-Thought (CoT). We report AUROC, AUPRC, and Top-50% ACC following the setting of `Section 4.1.1` in our paper.
> + **Code Generation:** We evaluate on **HumanEval** [2], a standard benchmark for code generation. We report AUROC, AUPRC, and Top-50% ACC.
> + **Truthfulness:** We evaluate on **FactScore** [3], a widely used benchmark for factual correctness. We use *GPT-5-mini* to decompose model outputs into atomic facts and judge them. Here we report Top-50% Precision (FactScore) when ranking responses by uncertainty.
>
> Given the increased difficulty of the newly added tasks and the model-diversity concern raised in `W3`, we perform all additional evaluations using a different base LLM, *Qwen-2.5-3B-Instruct.* The results are summarized in Table A.
>
> **Table A. Performance of Uncertainty Estimation Methods for Incorrect Reasoning Detection on 5 Long-Form Generation Tasks with Qwen-2.5-3B-Instruct.**
>
> | Method | Zebra Puzzles | | | Leg Counting | | | Color Cube | | | HumanEval | | | FactScore |
> | - | - | - | - | - | - | - | - | - | - | - | - | - | - |
> | | AUROC | AUPRC | ACC | AUROC | AUPRC | ACC | AUROC | AUPRC          | ACC            | AUROC          | AUPRC          | ACC            | Precision      |
> | PE               | 44.70±0.00     | 21.76±0.00     | 24.00±0.00     | 38.87±0.25     | 27.44±0.05     | 26.00±0.00     | 46.73±0.00     | 28.75±0.00     | 22.00±0.00     | 60.52±0.00     | 81.45±0.00     | 78.05±0.00     | 41.86±0.02     |
> | LL               | 42.15±0.00     | 21.02±0.00     | 22.00±0.00     | 37.92±0.43     | 27.10±0.13     | 32.67±1.15     | 46.36±0.00     | 26.01±0.00     | 24.00±0.00     | 56.42±0.00     | 79.48±0.00     | 75.61±0.00     | 45.03±0.38     |
> | Self-Certainty   | 47.77±0.00     | 22.95±0.00     | 30.00±0.00     | 50.46±0.30     | 32.88±0.37     | 34.00±0.00     | 50.62±0.00     | 30.80±0.00     | 28.00±0.00     | 61.48±0.00     | 80.27±0.00     | 80.49±0.00     | 40.10±0.38     |
> | DeepConf         | 42.41±0.00     | 21.07±0.00     | 26.00±0.00     | 42.48±0.71     | 28.96±0.32     | 30.00±0.00     | 49.01±0.00     | 24.41±0.00     | 24.00±0.00     | **63.60±0.00**     | **82.75±0.00** | 80.49±0.00     | 41.96±0.49     |
> | TokUR (TU, Ours) | 71.38±0.29     | 41.42±0.38     | **39.33±1.15** | 69.58±2.05     | 48.16±0.99     | 48.67±1.15     | 58.33±2.86     | 31.14±1.56   | 31.33±1.15   | 63.15±0.82     | 81.59±0.41     | **80.90±0.70** | 60.35±0.74     |
> | TokUR (AU, Ours) | **71.66±0.35** | **41.71±0.44** | **39.33±1.15** | 69.51±1.82     | 47.69±0.70     | 48.67±1.15     | 56.57±2.88     | 30.02±1.36     | 28.00±2.00     | 60.50±0.80     | 81.75±0.41     | **80.90±1.41** | **61.53±0.58** |
> | TokUR (EU, Ours) | 71.28±0.38     | 41.20±0.76     | 37.33±2.31     | **69.66±2.51** | **51.27±2.07** | **50.67±2.31** | **60.08±1.98** | **33.71±1.01** | **32.00±2.00** | 60.64±0.77 | 80.75±0.48     | 80.49±0.00     | 45.26±0.54     |
>
>
> Across all non-math tasks, **our TokUR consistently ranks as the best or second-best method, confirming that it generalizes effectively to diverse long-form generation settings and retains strong performance beyond the mathematical reasoning domain.**
>
> We will include the discussion and results above and update the paper as suggested.

---

> > ### Author Response · Authors · 2025-11-21
> > **Official Rebuttal by Authors (2/4)**
> >
> > **W2.1 [Cost Analysis] "While the cost is the common challenge of uncertainty estimation, it seems that there is no thorough cost analysis in the paper, compared to prior methods like query-level uncertainty estimation."**
> >
> > Thank you for mentioning this. As a Bayesian approach that relies on Monte Carlo sampling for expectation estimation, **TokUR incurs approximately $N\times$** the computational cost of standard single-sample LLM inference, where $N$ denotes the number of stochastic forward passes at test time. This overhead is inherent to **all Bayesian Model Averaging (BMA) approaches**, and improving test-time efficiency is **NOT** the primary focus of this work.
> >
> > Fortunately, as discussed in `Section 3.3`, TokUR's design incorporates `a low-rank noise matrix` to minimize the additional sampling cost. **Empirically, we find that TokUR delivers strong uncertainty-estimation performance with a small sampling budget (e.g., $N=2$ to $N=5$)**. Moreover, **our implementation is fully compatible with vLLM,** enabling efficient batched sampling and low-overhead deployment.
> >
> > Inspired by your comment and for completeness, Table B below summarizes the inference cost of all query-level uncertainty estimation for both our TokUR and the baselines adopted in our experiments.
> >
> > **Table B. Inference Cost of Query-Level Uncertainty Estimation for TokUR and Baselines.**
> >
> > | Method | \#Forward Passes per Query |
> > | ------ | ------------------------------------------------------------ |
> > | SE     | $\times (N + M)$, where $M$ is the cost of an additional NLI model |
> > | SAR    | $\times (N + M)$, where $M$ is the cost of an additional semantic-similarity model |
> > | INSIDE | $\times N$ |
> > | **TokUR (Ours)** | $\times N$ |
> >
> > Note that these baselines (e.g., SE and SAR) require generating multiple *full* responses per query, often with additional auxiliary models, leading to **equal or higher total cost than TokUR**. Thus, although TokUR inherits the sampling overhead typical of Bayesian approaches, its overall cost remains competitive compared to existing query-level uncertainty estimation techniques.
> >
> > We will include the discussion above and revise the paper accordingly as suggested.
> >
> >
> > **W2.2 [Cost Analysis] "Also, how does the estimation technique used in Section 3.3 help reduce cost?"**
> >
> > `Section 3.3` introduces a **low-rank weight perturbation** strategy that reduces the cost of Bayesian sampling compared to full-rank perturbation.
> >
> > Instead of drawing full-rank Gaussian perturbations over all model parameters, which would require sampling over millions or billions of dimensions, TokUR applies perturbations **only within a small low-rank subspace** obtained via SVD of each weight matrix. Concretely, for a rank-$r$ weight matrix $W_0$, we perturb only the top-$r'$ singular directions ($r' \ll r$) and construct the perturbed weight as
> > $$
> > W = W_0 + U' \epsilon^\top ,
> > $$
> > where $U'$ contains only the top-$r'$ columns of $U$, and $\epsilon$ is a small $n \times r'$ Gaussian noise matrix.
> > This means the stochastic dimension of the perturbation is reduced from **$O(nr)$** to **$O(n r')$**, cutting the sampling cost by a factor of $\frac{r'}{r}$.

---

> > > ### Author Response · Authors · 2025-11-21
> > > **Official Rebuttal by Authors (3/4)**
> > >
> > > **W3. [Model Diverisity] "Evaluating two models from the same Llama family seems insufficient to support the correlation between accuracy and token-level uncertainty estimation. Including a few more models from different family and sizes is encouraged to reduce bias in experiments and to strengthen the claim."**
> > >
> > > Thank you for mentioning this. Following your suggestion, we expanded our evaluation to include models from additional families and sizes. As shown in our response to **W1**, we added comprehensive experiments on *Qwen-2.5-3B-Instruct* across multiple *non-math* long-form generation tasks, demonstrating that TokUR maintains strong uncertainty–performance alignment beyond the Llama family.
> > >
> > >
> > > Furthermore, we extended the current the experiments in `Table 1` of the paper to **two additional base LLMs** from the *Qwen* family: *Qwen-2.5-3B-Instruct* and *Qwen-2.5-7B-Instruct*. The results are summarized in Tables C.1 and C.2 as below. (Due to time and computational constraints, our experiments on additional models from the *Gemma* family are not yet fully completed. We will post the new results once they become available.)
> > >
> > >
> > > **Table C.1. Performance of Uncertainty Estimation Methods for Incorrect Reasoning Detection on Math-Reasoning Tasks with Qwen-2.5-3B-Instruct.**
> > >
> > > | Method           | MATH500 AUROC  | MATH500 AUPRC  | MATH500 ACC    | GSM8K AUROC    | GSM8K AUPRC    | GSM8K ACC      | DeepScaleR AUROC | DeepScaleR AUPRC | DeepScaleR ACC |
> > > | ---------------- | -------------- | -------------- | -------------- | -------------- | -------------- | -------------- | ---------------- | ---------------- | -------------- |
> > > | PE               | 68.94±0.27     | 77.30±0.62     | 76.80±0.40     | 76.44±0.10     | 95.06±0.01     | 94.46±0.00     | 66.57±0.07       | 46.44±0.02       | 48.09±0.02     |
> > > | LL               | 68.42±0.29     | 76.82±0.79     | 76.13±0.23     | 75.54±0.12     | 94.93±0.01     | 94.92±0.00     | 65.66±0.06       | 45.43±0.02       | 47.37±0.02     |
> > > | Self-Certainty   | 74.95±0.40     | 84.76±0.40     | 81.33±0.61     | 77.22±0.00     | 95.66±0.02     | 94.77±0.00     | 74.22±0.08       | 62.55±0.02       | 52.00±0.00     |
> > > | DeepConf         | 74.39±0.33     | 83.81±0.39     | 80.53±0.61     | 78.06±0.01     | 95.63±0.03     | 95.69±0.00     | 73.85±0.07       | 60.75±0.02       | 51.93±0.02     |
> > > | TokUR (TU, Ours) | 81.48±0.09     | 87.42±0.09     | 86.67±0.23     | 77.66±0.40     | 95.46±0.12     | 96.05±0.18     | 76.26±0.02       | 65.47±0.07       | 53.52±0.14     |
> > > | TokUR (AU, Ours) | **82.55±0.05** | **88.30±0.15** | **87.33±0.23** | **78.81±0.31** | **95.76±0.16** | **96.15±0.31** | **77.52±0.03**   | **67.30±0.08**   | **54.61±0.27** |
> > > | TokUR (EU, Ours) | 78.96±0.13     | 85.37±0.24     | 84.27±0.46     | 70.75±0.81     | 93.60±0.21     | 94.26±0.44     | 73.33±0.02       | 61.08±0.13       | 51.35±0.10     |
> > >
> > >
> > > **Table C.2. Performance of Uncertainty Estimation Methods for Incorrect Reasoning Detection, evaluated on math-reasoning tasks, with Qwen-2.5-7B-Instruct.**
> > >
> > > | Method           | MATH500 AUROC  | MATH500 AUPRC  | MATH500 ACC    | GSM8K AUROC    | GSM8K AUPRC    | GSM8K ACC      | DeepScaleR AUROC | DeepScaleR AUPRC | DeepScaleR ACC |
> > > | ---------------- | -------------- | -------------- | -------------- | -------------- | -------------- | -------------- | ---------------- | ---------------- | -------------- |
> > > | PE               | 62.51±0.37     | 80.29±0.45     | 82.00±0.40     | 77.40±0.19     | 97.35±0.01     | 97.23±0.00     | 60.55±0.09       | 50.61±0.10       | 52.64±0.14     |
> > > | LL               | 62.19±0.39     | 80.19±0.46     | 82.13±0.61     | 75.37±0.22     | 97.02±0.00     | 97.08±0.00     | 60.23±0.10       | 50.27±0.11       | 52.43±0.14     |
> > > | Self-Certainty   | 69.54±0.06     | 86.24±0.10     | 85.60±0.40     | 80.92±0.15     | 97.81±0.01     | 97.69±0.00     | 66.31±0.07       | 58.18±0.10       | 56.60±0.08     |
> > > | DeepConf         | 66.86±0.06     | 83.85±0.03     | 84.53±0.23     | **81.94±0.15** | **97.88±0.00** | **98.31±0.00** | 64.91±0.05       | 56.71±0.08       | 55.83±0.08     |
> > > | TokUR (TU, Ours) | 84.61±0.62     | 93.43±0.25     | **94.00±0.40** | 80.15±0.12     | 97.36±0.11     | 97.69±0.00     | 76.05±0.12       | 71.43±0.31       | 63.89±0.33     |
> > > | TokUR (AU, Ours) | **84.74±0.52** | **93.65±0.17** | **94.00±0.40** | 80.70±0.10     | 97.44±0.14     | 97.79±0.09     | **76.39±0.10**   | **71.73±0.31**   | **64.24±0.30** |
> > > | TokUR (EU, Ours) | 81.87±1.02     | 91.68±0.35     | 92.00±1.06     | 74.95±0.40     | 96.61±0.08     | 97.23±0.15     | 73.55±0.21       | 68.91±0.48       | 61.47±0.13     |
> > >
> > >
> > > Overall, these additional experiments significantly improve model diversity and reinforce the claim that TokUR’s token-level uncertainty estimation correlates robustly with model accuracy across different LLM families and sizes.

---

> > > > ### Author Response · Authors · 2025-11-21
> > > > **Official Rebuttal by Authors (4/4)**
> > > >
> > > > **Q1. "It would be helpful to analyze the length-robustness of your token-level uncertainty estimates. As responses grow longer, how do the estimation errors (or bounds) evolve?"**
> > > >
> > > > This is a good point. Following your suggestion, to examine how well our TokUR works as generation length increases, we conducted a quantile-based analysis using the data in `Figure 1`. Specifically, we partitioned all responses into five length quantiles (0–20%, 20–40%, $\dots$, 80–100%) and computed AUROC for TokUR (AU/TU/EU) and LL (as a reference baseline) within each bin. The results are summarized in Table D.
> > > >
> > > > **Table D. Correlation between Response Length and Our TokUR's AUROC (%).**
> > > >
> > > > | Quantile Range | Avg. Length | LL    | TokUR(AU, Ours) | TokUR(TU, Ours) | TokUR(EU, Ours) |
> > > > | -------------- | ---------- | ----- | --------- | --------- | --------- |
> > > > | 0–20%          | 181.05     | 65.86 | 68.05     | 68.15     | 67.66     |
> > > > | 20–40%         | 286.44     | 61.87 | 63.50     | 63.63     | 67.28     |
> > > > | 40–60%         | 393.83     | 68.80 | 69.90     | 69.99     | 68.39     |
> > > > | 60–80%         | 609.26     | 63.00 | 66.28     | 66.37     | 67.99     |
> > > > | 80–100%        | 1732.22    | 42.77 | 54.91     | 55.00     | 60.15     |
> > > >
> > > > These results show that **TokUR is highly robust against the response length.** Across the first four quantiles (containing 80% of the data points), where the average lengths range from **181 to 609 tokens**, TokUR enjoys **consistent AUROC between 63–70** for all uncertainty components (AU, TU, and EU), with no monotonic degradation or systematic trend. This indicates that TokUR does **not suffer from length-induced bias**, and that its estimation quality remains stable even as sequences become substantially longer ($\approx 3 \times$ increase from the 1st to the 4th quantile).
> > > >
> > > > Moreover, even in the extreme 80–100% quantile (average token length >1700), where all methods naturally degrade due to increased reasoning complexity, TokUR still retains **significantly stronger discriminative ability ($\approx$55%-60% AUROC)** compared to LL's **42.8 AUROC**.
> > > >
> > > > These results collectively confirm that compared to baselines, **TokUR does not significantly suffer from length-induced bias and provides substantially more robust uncertainty estimates.** We will include this analysis and the table above in our revision as suggested.
> > > >
> > > > **References**
> > > > + [1] Stojanovski, Zafir, Oliver Stanley, Joe Sharratt, Richard Jones, Abdulhakeem Adefioye, Jean Kaddour, and Andreas Köpf. "REASONING GYM: Reasoning Environments for Reinforcement Learning with Verifiable Rewards." arXiv preprint arXiv:2505.24760 (2025).
> > > > + [2] Chen, Mark. "Evaluating large language models trained on code." arXiv preprint arXiv:2107.03374 (2021).
> > > > + [3] Min, Sewon, Kalpesh Krishna, Xinxi Lyu, Mike Lewis, Wen-tau Yih, Pang Koh, Mohit Iyyer, Luke Zettlemoyer, and Hannaneh Hajishirzi. "Factscore: Fine-grained atomic evaluation of factual precision in long form text generation." In Proceedings of the 2023 Conference on Empirical Methods in Natural Language Processing, pp. 12076-12100. 2023.

---

> > > > > ### Comment · Reviewer_9t1g · 2025-11-25
> > > > >
> > > > > I thank the authors for their effort in fully addressing my concerns. I am happy to raise the score accordingly.

---

> > > > > > ### Author Response · Authors · 2025-11-26
> > > > > > **Thank you.**
> > > > > >
> > > > > > Thank you very much for the encouraging comments and for acknowledging our contribution. We are glad that our response has fully addressed your concerns. We will make sure to include the discussion above in our next revision.

---

### Author Response · Authors · 2025-11-24
**Summary of Author Response**

We sincerely thank all reviewers for their time and constructive feedback. We are glad that the reviewers find

+ **our problem studied** as addressing `"a key practical need for safe and reliable LLM deployment"` where confidence calibration is critical [*ZNDU*], with `"clear positioning vs. prior work"` [*9t1g*];
+ **our proposed uncertainty estimation method** as `"elegant"` [*ZNDU*], `"novel"` [*YhsK*], with `"strong theoretical ground"` and `"thoughtfully developed"` foundations [*9t1g*].
+ **our experiments** as `"fairly comprehensive"` [*YhsK*];
+ **our paper** as `"readable and well organized"` with `"clear notations"` [*9t1g*], containing `"well-explained"` `"technical concepts"` [*ZNDU*], and overall `"clearly written and easy to follow"` [*9t1g, YhsK*].


Regarding the raised concerns, we have addressed them in the rebuttal with new analyses and supporting evidence:

+ Reviewer 9t1g: We added a comprehensive set of new experiments demonstrating the **scalability of TokUR** across **three additional task types** (logical reasoning, coding, and truthful response generation), **two additional base LLMs,** and included an additional analysis showing **its consistency across varying generation lengths.**
+ Reviewer ZNDU: We expanded our experimental evaluation to include **the three additional task types noted above** and provided further **clarification on TokUR’s lower cost overhead** relative to other uncertainty-estimation baselines.
+ Reviewer YhsK: We **clarified the respective roles of EU and AU** within TokUR, **explained our existing definitions of query-level and response-level uncertainty,** and provided a detailed explanation showing that **weight perturbations are used exclusively for uncertainty estimation** and **never affect the base model’s generation.**

We will integrate these clarifications, ablations, and results into the revision, and will release all code and supplementary materials for full reproducibility.

We also thank the AC and SAC for their guidance and oversight throughout the review process.

---

> ### Author Response · Authors · 2025-11-29
> **Manuscript Updated**
>
> Thank you again to all reviewers for their time and service. We are especially grateful to `Reviewer 9t1g` for thoughtfully considering our rebuttal and **raising the score from 4 to 6 on Nov. 25, well before** the major information-leak incident on **Nov. 27**. For transparency, **our scores before the rollback were [6, 6, 4].**
>
> As summarized above, we have incorporated all reviewer suggestions into the revised manuscript, with changes highlighted in blue.
>
> We also sincerely thank the AC and SAC for their continued guidance and oversight throughout the review process.
>
> TokUR Authors

---

### Meta-Review · Area_Chair_wzHB · 2026-01-08

**Summary:**

The paper proposes a Token-level Uncertainty estimation framework for Reasoning (TokUR) that enables LLMs to self-assess and self-improve their responses in mathematical reasoning. Reviewers generally find the topic meaningful and the approach theoretically well grounded, describing the method as effective and elegant.

Most reviewers’ main concern is that the experimental evaluation is incomplete—for example, the paper lacks results on additional datasets/tasks, does not sufficiently test the method across different model backbones, and provides limited analysis of computational cost/overhead. Reviewer YhsK additionally points out potential issues in the theoretical development, noting possible inconsistencies between the theory and the empirical results.

**Reviewer Concerns:**

The authors added substantial additional experiments in the rebuttal, which I believe largely addresses the reviewers’ concerns about the empirical evaluation. However, I am not sure whether the authors’ clarification and explanation of the theory will be sufficient to convince Reviewer YhsK.

**Reviewer Scores:**

I think Reviewer 9t1g to increase their score from 4 to 6.

---

### Decision · Program_Chairs · 2026-01-26

Accept (Poster)